# INFERENCE-TIME UNLEARNING VIA ADAPTIVE OUTPUT REGULATION

## ABSTRACT

Large Language Models (LLMs) have demonstrated strong capabilities in memorizing vast amounts of knowledge across diverse domains. However, the ability to selectively forget specific knowledge is critical for ensuring the safety and compliance of deployed models. Existing unlearning efforts typically fine-tune the model with resources such as forget data, retain data, and a calibration model. These additional gradient steps blur the decision boundary between forget and retain knowledge, often resulting in degraded overall performance. To avoid the negative impact of fine-tuning, it would be better to achieve *approximate unlearning at inference time*, where the model is dynamically guarded against generating responses related to the forget target without retraining or damaging its fluency. Current training-free approaches, though avoiding retraining, often suffer from incomplete or superficial forgetting. To this end, we introduce **GUARD**, an inference-time unlearning approach via adaptive output regulation to mitigate this problem without retraining or compromising fluency, which first employ a prompt classifier to detect unlearning targets and extract the corresponding forbidden tokens. We then dynamically penalize and filter candidate tokens during generation through a combination of token matching and semantic matching, thereby preventing the model from leaking the forgotten content. Experimental results on copyright-content unlearning tasks over the Harry Potter dataset and the MUSE benchmark, as well as entity unlearning tasks on the TOFU dataset, demonstrate that **GUARD** achieves strong forget quality across various tasks while causing almost no degradation to the LLM's general capabilities, striking an excellent trade-off between forgetting and utility.

## 1 INTRODUCTION

The rapid development of large language models (LLMs) has garnered widespread attention from academia and industry, driving significant progress across diverse fields (Achiam et al., 2023; Team et al., 2023; Touvron et al., 2023; Guo et al., 2025; Singhal et al., 2023; Taylor et al., 2022). However, additional challenges are also observed to ensure the safe and trustworthy deployment of LLMs, i.e., privacy protection (Staab et al., 2023; Mireshghallah et al., 2023; Das et al., 2025), copyright compliance (Karamolegkou et al., 2023; Grynbaum & Mac, 2023; Chu et al., 2024), content reliability (Harandizadeh et al., 2024; Zhang et al., 2023; Chua et al., 2024), etc. During training, LLMs may accidentally memorize sensitive personal data or copyright-relevant material, leading to biased or inaccurate output and associated risks (Tirumala et al., 2022; Carlini et al., 2021; Barez et al., 2025). To mitigate these issues, regulations such as GDPR (European Union, 2016) require the deletion of specific data upon user request. Although retraining the pre-trained LLM is the most direct solution, its high computational cost has spurred the growth of LLM Unlearning (Cao & Yang, 2015; Jia et al., 2023; Fan et al., 2023; Liu et al., 2025; Xu, 2024; Wang et al., 2024; Yao et al., 2024b; Ding et al., 2024; Cha et al., 2024; Ramakrishna et al., 2025), which aims to remove training influences from the forget data while maintaining overall performance.

Existing LLM unlearning methods can be broadly categorized into fine-tuning-based and training-free approaches. Fine-tuning-based methods mitigate the influence of target information by fine-tuning the model on a small-scale forget data, with regularization on the retain data to prevent excessive forgetting of unrelated knowledge (Maini et al., 2024a; Wang et al., 2024; Zhang et al., 2024; Yao et al., 2024b; Huo et al., 2025). These methods require only minor parameter updates

Figure 1: Overview of **GUARD**: In Step 1, we use an MLP to determine whether the prompt belongs to the forget target; In Step 2, we retrieve the original answer from the forget data $D_f$ and extract the forbidden token, which consists of key phrases that should no longer appear in model outputs; In Step 3, we perform unlearning by dynamically suppressing target tokens during generation using token-level hard matching and SBERT-based semantic matching.

without retraining from scratch. In contrast, training-free methods leverage in-context examples to guide the original LLM to forget specific information without modifying its parameters (Pawelczyk et al., 2023; Muresanu et al., 2024; Thaker et al., 2024). However, existing studies have shown that fine-tuning-based methods often lead to a decrease in model utility and cause catastrophic forgetting (Chen et al., 2025; Lynch et al., 2024). Although the current goal of unlearning is to approximate the effect of retraining, existing training-free approaches often could not meet this level of performance (Liu et al., 2025). These challenges underscore that balancing effective forgetting while preserving overall model performance remains a central difficulty in LLM unlearning research.

In this work, we introduce an inference-time unlearning approach via adaptive output regulation to mitigate the performance degradation and boundary blurring caused by fine-tuning, without retraining or compromising fluency. Specifically, we present **G**eneration-time **U**nlearning via **A**daptive **R**estriction and **D**etection (**GUARD**). As illustrated in Figure 1, **GUARD** consists of three steps: In Step 1, we use a simple MLP, which takes the pre-computed embedding of the prompt as input, to classify whether the input prompt belongs to the forget target or not. In Step 2, for prompts that are categorized as forget target, we retrieve the original answer and extract the forbidden token. In Step 3, we apply a token-level hard matching strategy to identify and block forbidden token sequences during generation, and combine it with an SBERT-based (Reimers & Gurevych, 2019) semantic soft matching strategy to dynamically penalize and filter candidate tokens, thereby preventing the model from recalling forgotten content.

Our contributions are mainly two folds:

- We introduce **G**eneration-time **U**nlearning via **A**daptive **R**estriction and **D**etection (**GUARD**), a generation-time approach to achieve approximate LLM unlearning without retraining or fine-tuning. The design of **GUARD** **does not** touch on updates of LLM parameters, ensuring the fluency of the generated language after unlearning, and maintaining performance as close as possible to that of the retained model, without causing catastrophic forgetting.

- Extensive experiments on three LLM Unlearning tasks, including unlearning copyright content from the Harry Potter dataset and the MUSE benchmark, as well as entity unlearning on the TOFU dataset, demonstrate the **superior performance** of our method, maintaining the model utility to the largest content while ensuring satisfying forget quality.

## 2 RELATED WORK

**Fine-tuning-based LLM unlearning methods.** Fine-tuning-based methods update model parameters via reverse gradient optimization. GA (Bourtoule et al., 2020) removes specific memories by maximizing the loss w.r.t. the forget data. Later, GD (Wang et al., 2023) expands GA by incorporating the retain data to balance the forget quality and model utility, preserving overall model performance. Further studies propose customized loss functions, such as PD Loss (Chen et al., 2025) to mitigate over-forgetting, or composite objectives that combine standard losses with regu-

larization terms (Yao et al., 2024b). Some methods fine-tune models using counterfactual answers (Gu et al., 2024), refusal responses (Maini et al., 2024a), or domain-consistent alternatives (Mekala et al., 2024) to enforce unlearning. In addition, reference models guide optimization via KL minimization (Yao et al., 2024a), NPO (Zhang et al., 2024), DPO (Rafailov et al., 2023), and KTO (Ethayarajh et al., 2024), enabling finer control over output distributions during fine-tuning.

**Training-free LLM unlearning methods.** Training-free methods typically do not modify the model parameters but instead achieve unlearning by altering the input prompts to steer the model away from its original output distribution (Pawelczyk et al., 2023; Muresanu et al., 2024; Thaker et al., 2024; Gao et al., 2024). ECO Prompt (Liu et al., 2024) uses a lightweight classifier to identify inputs requiring unlearning, and then applies embedded perturbations to disrupt the prompts, thereby guiding the model's output toward a "never-seen" state. Soft Prompt Unlearning (Bhaila et al., 2024) employs learnable soft prompts within the context to dilute target memories, enabling rapid unlearning without parameter updates. Proactive Privacy Amnesia (Kuo et al., 2025) integrates a PII detector with a multi-round adaptive refusal strategy, significantly reducing privacy leakage while largely preserving model utility.

## 3 PRELIMINARIES

### 3.1 DATASET SETUP AND NOTATION

We consider a standard machine unlearning setup, where the full training dataset is denoted as $D = \{z_i = (\mathbf{x}_i, y_i)\}_{i=1}^N$, where $\mathbf{x}_i$ is the input data and $y_i$ denotes the corresponding labels. The dataset is divided into three disjoint subsets: a forget set $D_f$, a retain set $D_r$, and optionally, an auxiliary generalization set $D_g$, which is drawn from an out-of-distribution source. A learning algorithm $A$ maps the dataset $D$ to a parameterized model $\theta = A(D)$.

The following notations distinguish different models derived from the dataset: $\theta_o = A(D)$ is the original model trained on the full dataset. $\theta_r = A(D_r)$ denotes the retained model, which is trained from scratch on the retain set $D_r$, excluding $D_f$. Finally, $\theta_u$ refers to the unlearned model, which is produced by an unlearning algorithm $U$, ideally approximating $\theta_r$ without requiring retraining.

### 3.2 FINE-TUNING-BASED UNLEARNING

Many existing unlearning methods (Yao et al., 2024b; Maini et al., 2024a; Wang et al., 2024; Zhang et al., 2024; Chen et al., 2025; Chen & Yang, 2023) approach the problem by formulating it as a regularized fine-tuning process, optimizing an objective of the following form:

$$\mathcal{L}_{\text{total}} = \lambda_1 \mathcal{L}_{\text{forget}} + \lambda_2 \mathcal{L}_{\text{retain}} + \lambda_3 \mathcal{L}_{\text{custom}}, \tag{1}$$

where $\mathcal{L}_{\text{forget}}$ encourages forgetting, often through gradient ascent or loss maximization on $D_f$, $\mathcal{L}_{\text{retain}}$ ensures that the model preserves performance on $D_r$, and $\mathcal{L}_{\text{custom}}$ provides greater flexibility and customization in the unlearning process. However, these approaches typically rely on directly modifying the model parameters, which may risk catastrophic forgetting.

### 3.3 GENERATION-TIME UNLEARNING

In contrast to traditional fine-tuning-based methods, our approach performs unlearning directly during generation time, without modifying the original model parameters. Given a fixed, fully-trained model $\theta_o$, we construct an unlearned model $\theta_u$ by applying an adaptive perturbation mechanism in the output space. Specifically, for each input $\mathbf{x}$ that corresponds to a forgetting target, we define:

$$h(\mathbf{x}; \theta_u) = \text{Unlearn}(h(\mathbf{x}; \theta_o)), \tag{2}$$

where $h(\mathbf{x}; \theta_o)$ denotes the logits or soft predictions from model $\theta_o$. The key objective is to selectively suppress the memorization of content associated with the forget set $D_f$, while preserving similarity to the retrained model $\theta_r$ on the retain set $D_r$, and maintaining generalization performance on $D_g$.

# 4 METHOD

## 4.1 METHOD OVERVIEW

When certain samples need to be unlearned, traditional approaches typically rely on fine-tuning, which often introduces challenges, most notably, catastrophic forgetting that can compromise the overall model utility. In this section, we introduce **G**eneration-time **U**nlearning via **A**daptive **R**estriction and **D**etection (**GUARD**), a generation-time unlearning framework designed to prevent large language models from reproducing sensitive information marked for forgetting, without harming the model's general capabilities. Our framework consists of three main components:

- **Prompt classification:** We first train a prompt classifier to determine whether a given input query corresponds to a forgetting target;
- **Forbidden token extraction:** For inputs classified as forget queries, we retrieve the most semantically similar question from forget data $D_f$ and extract the corresponding forbidden token from its associated answer, which serves as the content to be suppressed;
- **Controlled generation:** During generation, we employ a beam search strategy, enhanced by a token-level hard matching and a semantic similarity detector based on Sentence-BERT (SBERT)[1] (Reimers & Gurevych, 2019). This enables dynamic penalization and filtering candidate tokens at each decoding step, thereby effectively preventing the model from recalling forgotten content.

## 4.2 PROMPT CLASSIFICATION

The first component of our framework focuses on **identifying whether a given prompt should be subject to unlearning**. To achieve this, we train a binary classifier that predicts whether an input prompt $\mathbf{x}$ belongs to the forget target or not. Instead of directly training a model, we adopt a two-stage approach: we first use a frozen LLM (which will later be unlearned) to extract semantic representations for each prompt, and then train a lightweight classifier based on these embeddings.

Formally, we denote by $\mathbf{z}_i \in \mathbb{R}^d$ the semantic embedding of the $i$-th prompt, obtained by averaging the hidden states from the penultimate layer of a frozen causal LLM as follows:

$$\mathbf{z}_i = \frac{1}{L_i} \sum\nolimits_{,j=1}^{L_i} \mathbf{h}_{i,j}^{(l)} \cdot \mathbf{m}_{i,j}, \tag{3}$$

where $\mathbf{h}_{i,j}^{(l)}$ denotes the hidden state at position $j$ from the $l$-th layer, $\mathbf{m}_{i,j} \in \{0, 1\}$ is the attention mask, and $L_i = \sum_j \mathbf{m}_{i,j}$ is the actual length of the input. These embeddings $\mathbf{z}_i$ are then used to train a binary classifier $C(\cdot)$, implemented as an MLP, which outputs the predicted probability of the prompt belonging to the forget class:

$$p_C(f \mid \mathbf{z}_i) = \text{Softmax}(\mathbf{W}\mathbf{z}_i + \mathbf{b})_f, \tag{4}$$

where $\mathbf{W}$ and $\mathbf{b}$ are the learnable weight matrix and bias vector of the MLP output layer, and $\text{Softmax}(\cdot)_f$ denotes the probability assigned to the forget class. If a prompt is classified as forget, we proceed to the next stage. We provide further details on the training process, along with comprehensive robustness evaluations of the classifier under a wide range of jailbreak attempts and contextually distracting conditions, in Appendix B.

## 4.3 FORBIDDEN TOKEN EXTRACTION

Once an input query is classified as a forget prompt, we **retrieve the most relevant QA pair from the forget set** $D_f$. Let $\mathcal{A} = \{A_1, A_2, \ldots, A_M\}$ denote the set of answers extracted from $\mathcal{D}_f$, where each answer $A_i$ may contain sensitive information that should be forgotten.

To identify the most relevant forgetting answer $A^*$ for query $\mathbf{x}$, we adopt a semantic similarity-based retrieval strategy. Specifically, we compute the similarity between the query $\mathbf{x}$ and each candidate answer $A_i$ using a similarity function $\text{sim}(\mathbf{x}, A_i)$, and select the most similar one:

$$A^* = \arg \max_{A_i \in \mathcal{A}} \text{sim}(\mathbf{x}, A_i). \tag{5}$$

---

[1]sentence-transformers/all-MiniLM-L6-v2

The similarity function $\text{sim}(\cdot, \cdot)$ is implemented using SBERT (Reimers & Gurevych, 2019), which encodes both the input query and the candidate answers into dense embeddings and computes their cosine similarity. Here, we only persist the semantic embeddings of the forget set $D_f$. Details of the retrieval experiments can be found in Appendix C.

Once the most relevant answer $A^*$ is retrieved, we proceed to extract its sensitive textual fragments. We denote the extracted forbidden content as a set of text spans:

$$\mathcal{F}(A^*) = \{f_1, f_2, \ldots, f_K\}. \tag{6}$$

These fragments serve as the target content to be blocked in the subsequent generation stage. The method for extracting forbidden token from the answer is described in Appendix E.2. A comparison of different forbidden token extraction methods is provided in Sec.5.5.

At this point, we have obtained the forbidden token set $\mathcal{F}(A^*)$ associated with the current query, which will be used in the generation phase as a control signal to penalize candidate outputs that may reveal forgotten content, thus enabling the next stage of generation-time control.

## 4.4 Controlled Generation

During generation, we adopt a beam search strategy to iteratively expand candidate sequences while applying dynamic filtering and penalization at each time step to prevent the model from generating forget data related content. Formally, let the current generated token sequence be:

$$T_{1:n} = [t_1, t_2, \ldots, t_n], \tag{7}$$

we sample multiple top-ranked candidate tokens $t_{n+1}$ from the model's predictive distribution, and extend each candidate by appending it to the current prefix $T_{1:n}$. To ensure that sensitive content is not produced, we impose two types of penalization on the expanded candidates: token-level hard matching and SBERT-based soft semantic matching.

**Token-level hard matching.** To perform token-level hard matching, we construct a trie data structure containing a collection of forbidden sequences (i.e., tokenized sensitive phrases that must be forgotten). This structure enables efficient suffix matching on the generated sequence. At each generation step, given an extended candidate sequence $T_{1:n+1}$, we check whether its suffix matches any forbidden subsequence $f_k \in \mathcal{F}$. If a complete match is found or the matched length exceeds a predefined threshold $\beta$, we assign an infinite penalty to prune the candidate; otherwise, a penalty proportional to the match length is applied. The penalty function is defined as:

$$\mathcal{P}_{\text{token}}(T_{1:n+1}) = \begin{cases} \infty, & \text{if } \text{suffix}(T_{1:n+1}) \in \{f_k\}; \\ \alpha_{\text{token}} \cdot L_{\text{match}}, & \text{if } L_{\text{match}} < \beta; \\ 0, & \text{otherwise}, \end{cases} \tag{8}$$

where $L_{\text{match}}$ is the length of the longest matched suffix, $\alpha_{\text{token}}$ is a scaling factor, and we set $\beta = 1$ so that any nonzero match incurs an infinite penalty.

**SBERT-based soft semantic matching.** To go beyond exact matching, we use SBERT to compute the semantic similarity between the last generated word $w_{\text{last}}$ in $T_{1:n+1}$ and each forbidden token $f_k \in \mathcal{F}$. Let $s = \max_{f_k} \text{sim}(w_{\text{last}}, f_k)$, where $\text{sim}(\cdot, \cdot)$ is cosine similarity between SBERT embeddings. A hard penalty is applied if $s \geq \delta$; otherwise, a soft penalty scaled by $\alpha_{\text{sbert}}$ is used:

$$\mathcal{P}_{\text{sbert}}(T_{1:n+1}) = \begin{cases} \infty, & s \geq \delta, \\ \alpha_{\text{sbert}} \, s, & \text{otherwise}, \end{cases} \tag{9}$$

We set $\delta = 0.5$, and study its effect in Appendix H.

**Total penalization and beam update.** At each decoding step, the total penalty for $T_{1:n+1}$ is computed as the sum of two components:

$$\mathcal{P}_{\text{total}}(T_{1:n+1}) = \mathcal{P}_{\text{token}}(T_{1:n+1}) + \mathcal{P}_{\text{sbert}}(T_{1:n+1}). \tag{10}$$

If $\mathcal{P}_{\text{total}} = \infty$, the candidate is immediately pruned. Otherwise, its total cost $\mathcal{C}(T_{1:n+1})$ is computed by adding the penalty to the negative log-likelihood of the next token:

Table 1: We evaluate our approach and baseline methods on 1% TOFU dataset using three base LLMs: Llama2-7B, Phi-1.5B, and OPT-2.7B. The metrics reported include Forget Quality (FQ), Model Utility (MU), ROUGE-L on the retain set (R-RL), and ROUGE-L on the forget set (F-RL). For comparison, results from the original LLM and the retain-tuned LLM are also provided. The top two performing methods are marked with blue.

| Base LLM | Llama2-7B | | | | Phi-1.5B | | | | OPT-2.7B | | | |
|---|---|---|---|---|---|---|---|---|---|---|---|---|
| Metric | FQ(↑) | MU(↑) | F-RL(↓) | R-RL(↑) | FQ(↑) | MU(↑) | F-RL(↓) | R-RL(↑) | FQ(↑) | MU(↑) | F-RL(↓) | R-RL(↑) |
| Original LLM | 4.4883e-06 | 0.6239 | 0.9851 | 0.9818 | 0.0013 | 0.5195 | 0.9607 | 0.9276 | 0.0013 | 0.5112 | 0.7537 | 0.8807 |
| Retained LLM | 1.0 | 0.6267 | 0.4080 | 0.9833 | 1.0 | 0.5233 | 0.4272 | 0.9269 | 1.0 | 0.5067 | 0.4217 | 0.7669 |
| GA | 0.0068 | 0.5990 | 0.4817 | 0.9204 | **0.0541** | 0.5058 | 0.4914 | 0.8012 | 0.0286 | 0.4717 | 0.5222 | 0.7789 |
| KL | 0.0030 | 0.5994 | 0.4922 | 0.9172 | **0.0541** | 0.5063 | 0.4958 | 0.8003 | 0.0541 | 0.4937 | 0.4799 | 0.7551 |
| GD | 0.0068 | 0.5998 | 0.4869 | 0.9182 | 0.0286 | 0.5117 | 0.4991 | 0.7959 | 0.0541 | 0.4846 | **0.4405** | 0.7595 |
| LLMU | 0.0030 | 0.5999 | 0.4891 | 0.9236 | 0.0143 | 0.5083 | 0.3380 | 0.7685 | **0.1649** | 0.0 | 0.0144 | 0.0119 |
| PO | 0.0030 | 0.6323 | 0.1752 | 0.9169 | **0.0541** | 0.5064 | 0.4958 | 0.8003 | 0.0068 | 0.4586 | 0.1350 | 0.6378 |
| DPO-RT | 0.0068 | 0.6322 | 0.2595 | 0.9091 | **0.0541** | 0.5012 | 0.2890 | 0.7302 | **0.1649** | 0.0 | 0.0010 | 0.0036 |
| NPO-RT | 0.0030 | 0.5994 | 0.5049 | 0.9270 | 0.0286 | 0.5092 | 0.4877 | 0.8210 | 0.0541 | 0.4938 | 0.4998 | 0.7718 |
| FLAT (Pearson) | **0.0541** | 0.6130 | **0.4508** | 0.9347 | 0.0286 | 0.5155 | **0.4716** | 0.8692 | 0.0541 | 0.4958 | 0.3892 | 0.7879 |
| ICUL | 0.0005 | **0.6239** | 0.4772 | **0.9818** | 0.0286 | **0.5195** | 0.0564 | **0.9276** | 0.0143 | **0.5112** | 0.0897 | **0.8807** |
| Output Filtering | 0.0002 | **0.6239** | 0.0 | **0.9818** | 2.1563e-05 | **0.5195** | 0.0 | **0.9276** | 6.5768e-05 | **0.5112** | 0.0 | **0.8807** |
| Prompt | 0.0005 | **0.6239** | 0.5915 | **0.9818** | 0.0143 | **0.5195** | 0.1136 | **0.9276** | 0.0143 | **0.5112** | 0.7636 | **0.8807** |
| **GUARD** | **0.1649** | **0.6239** | **0.3910** | **0.9818** | **0.1649** | **0.5195** | **0.4214** | **0.9276** | **0.4045** | **0.5112** | **0.4257** | **0.8807** |

$$\mathcal{C}\big(T_{1:n+1}\big) = -\log P(t_{n+1} \mid T_{1:n}) + \mathcal{P}_{\text{total}}\big(T_{1:n+1}\big). \tag{11}$$

All candidate extensions are ranked by their total cost $\mathcal{C}$, and the top candidates are retained for the next beam search iteration. If a sequence is penalized to $\infty$ at any step, it is discarded entirely. This ensures that sensitive content marked for unlearning is never produced during generation.

## 5 EXPERIMENT

In this section, we evaluate the proposed method against existing baseline approaches on three established LLM unlearning tasks. Specifically, we consider entity unlearning on the TOFU dataset (Maini et al., 2024b) (Sec.5.2), general unlearning capabilities assessed via the MUSE-News benchmark (Shi et al., 2024) (Sec.5.3) and copyright-based content unlearning using the Harry Potter (HP) Series Book dataset (Yao et al., 2024b) (Sec.5.4). In addition, we conduct ablation studies in Sec.5.5 to further investigate the impact, effectiveness, and sensitivity of our proposed components. Please refer to Appendix B.4 and C for detailed results on prompt classification and similarity retrieval.

### 5.1 BASELINE METHODS

We compare **GUARD** against a diverse set of unlearning baselines, grouped into four categories. **Gradient-based methods** include Gradient Ascent (GA) (Jang et al., 2022), GradDiff (GD) (Liu et al., 2022), KL minimization (KL) (Maini et al., 2024b), Large Language Model Unlearning (LLMU) (Yao et al., 2024b), and Mismatch (Liu et al., 2024). **Preference-based methods** include Preference Optimizatio (PO) (Maini et al., 2024b), Direct Preference Optimization (DPO) (Rafailov et al., 2023), Negative Preference Optimization (NPO) (Zhang et al., 2024), and FLAT (Wang et al., 2024). **Model editing methods** include Task Vectors (Ilharco et al., 2022) and Who's Harry Potter (WHP) (Eldan & Russinovich, 2023). **Training-free methods** include In-Context Unlearning (ICUL) (Pawelczyk et al., 2023), Output Filtering (Thaker et al., 2024), and Prompt-based strategies. Detailed descriptions of these methods are provided in Appendix D, and the corresponding experimental settings are summarized in Appendix E.1.

### 5.2 ENTITY UNLEARNING

**Experiment setup.** The TOFU dataset is a synthetic QA benchmark centered on author biographies. The objective is to assess whether an LLM, initially trained on the full dataset containing all authors, can selectively unlearn a specified subset (e.g., 1%) of samples, while preserving its knowledge of the remaining fictional individuals as well as general real-world information. Following the set up of (Wang et al., 2024), we use Llama2-7B (Touvron et al., 2023), Phi-1.5B (Li et al., 2023a), and OPT-2.7B (Zhang et al., 2022) as the base models for evaluation. In addition, we further conduct experiments using Falcon3-7B-Instruct (Team, 2024), Llama3.2-3B-Instruct (Grattafiori et al., 2024), and Qwen2.5-7B-Instruct (Yang et al., 2024). The additional results are presented in Appendix H.

Table 2: The performance on the MUSE benchmark is evaluated across four criteria. We emphasize results in blue when the unlearning algorithm meets the criterion, and in red when it does not. For the metrics on $D_f$, lower values are preferred, whereas for the metrics on $D_r$, higher values are desired. Regarding PrivLeak, the results should ideally be close to 0. Significant negative or positive values indicate potential privacy leakage. * indicates values sourced directly from Wang et al. (2024).

| | VerbMem on $D_f$ ($\downarrow$) | | KnowMem on $D_f$ ($\downarrow$) | | KnowMem on $D_r$ ($\uparrow$) | | PrivLeak |
|---|---|---|---|---|---|---|---|
| Original LLM | 58.4 | - | 63.9 | - | 55.2 | - | -99.8 |
| Retained LLM | 20.8 | - | 33.1 | - | 55.0 | - | 0.0 |
| Task Vectors* | 56.3 | (�’) | 63.7 | (✗) | 54.6 | (✔) | -99.8 |
| WHP* | 19.7 | (✔) | 21.2 | (✔) | 28.3 | (✔) | 109.6 |
| GA* | 0.0 | (✔) | 0.0 | (✔) | 0.0 | (✗) | 17.0 |
| GD* | 4.9 | (✔) | 27.5 | (✔) | 6.7 | (✔) | 109.4 |
| KL* | 27.4 | (✗) | 50.2 | (✗) | 44.8 | (✔) | -96.1 |
| NPO* | 0.0 | (✔) | 0.0 | (✔) | 0.0 | (✗) | 15.0 |
| NPO-RT* | 1.2 | (✔) | 54.6 | (✗) | 40.5 | (✔) | 105.8 |
| FLAT (Pearson)* | 1.6 | (✔) | 0.0 | (✔) | 0.2 | (✔) | 26.8 |
| ICUL | 10.7 | (✔) | 19.7 | (✔) | 55.2 | (✔) | -99.8 |
| Output Filtering | 1.1 | (✔) | 0.3 | (✔) | 55.2 | (✔) | -99.8 |
| Prompt | 15.4 | (✔) | 47.9 | (✗) | 55.2 | (✔) | -99.6 |
| **GUARD** | 4.3 | (✔) | 4.9 | (✔) | 55.2 | (✔) | 109.6 |

**Evaluation metrics.** To evaluate both forgetting effectiveness and model utility, we adopt two metrics from the TOFU benchmark: **Forget Quality (FQ)** and **Model Utility (MU)** (Maini et al., 2024a) . FQ is measured via the $p$-value of a Kolmogorov–Smirnov (KS) test comparing unlearned and retained model, a higher $p$-value indicates better forgetting. MU evaluates performance on retain data. We additionally report **ROUGE-L** scores on both forget and retain sets, noting that on the forget set, a ROUGE-L score closer to that of the retained model indicates more desirable unlearning behavior. Full metric details are provided in Appendix F.1.

**GUARD achieves good forget quality.** As shown in Table 1, our method achieves the best FQ performance across all three base models on the 1% dataset. Further, we provide evaluation results for the 5% and 10% datasets in Tables 10 and 11, where our method consistently demonstrates excellent forget quality in these scenarios as well. Moreover, **GUARD** consistently outperforms all training-free baseline methods across all splits. This demonstrates that existing prompt-based or template-based unlearning methods are insufficient to achieve satisfactory FQ, whereas our method enables the model to better approximate the distribution of the retained model.

**GUARD achieves the best trade-off.** Unlike most unlearning methods that risk catastrophic forgetting via fine-tuning, **GUARD causes no degradation in utility**. As shown in Tables 10 and 11, most of the baselines sacrifice utility for forgetting, reducing the MU to 0, while **GUARD** retains the same MU as the original model. Notably, across all splits, **GUARD** consistently ranks among the top two in terms of F-RL. This indicates that our method not only achieves strong forget quality, but also maintains high-quality generation that closely aligns with the performance of the retained model.

### 5.3 MUSE-NEWS UNLEARNING

**Experiment setup.** We evaluate our method on the MUSE-News benchmark (Shi et al., 2024), which is designed to simulate realistic unlearning scenarios on textual data. The MUSE-News dataset consists of BBC news articles (Li et al., 2023b) collected after August 2023, and is partitioned into three mutually disjoint subsets: a forget set containing the target data for removal, a retain set containing domain-relevant content to be preserved, and a holdout set for utility evaluation. For all experiments, we perform unlearning on the pretrained Llama2-7B (Touvron et al., 2023) model provided by the MUSE benchmark. Among the unlearning methods evaluated, prompt based method and **GUARD** are implemented by us, while the results of other baseline methods are taken from or reproduced according to their original implementations (Wang et al., 2024), following the same evaluation protocol as the MUSE benchmark.

**Evaluation metrics.** We evaluate our method using four metrics from the MUSE benchmark. ***VerbMem*** measures the model's ability to reproduce exact forgotten text, while ***KnowMem*** evaluates

Table 3: Performance of our method and the baseline methods on Harry Potter dataset using OPT-2.7B and Llama2-7B. The results for both models are shown, with best results across three main metrics highlighted in blue . The performance is evaluated using Forget Quality Gap (FQ Gap), perplexity (PPL), and average zero-shot accuracy (Avg. Acc.) across nine LLM benchmarks. * indicates values sourced directly from Wang et al. (2024).

| Base LLM | OPT-2.7B | | | Llama2-7B | | |
|---|---|---|---|---|---|---|
| Metric | FQ Gap(↓) | PPL(↓) | Avg. Acc.(↑) | FQ Gap(↓) | PPL(↓) | Avg. Acc.(↑) |
| Original LLM | 1.5346 | 15.6314 | 0.4762 | 3.6594 | 8.9524 | 0.5617 |
| Retained LLM | 0.0 | 14.3190 | 0.4686 | 0.0 | 8.7070 | 0.5599 |
| GA* | 2.7301 | 1.0984e71 | 0.3667 | 0.4587 | 47.2769 | 0.5088 |
| KL* | 2.7301 | 16.1592 | 0.4688 | 0.4225 | 9.4336 | 0.5509 |
| GD* | 2.3439 | 16.1972 | 0.4690 | 0.5304 | 9.1797 | 0.4902 |
| Mismatch* | 1.4042 | 15.7507 | 0.4679 | 0.4647 | 8.9906 | 0.5593 |
| LLMU* | 2.4639 | 15.8398 | 0.4656 | 0.1985 | 9.0530 | 0.5503 |
| PO* | 2.1601 | 14.8960 | 0.4583 | 0.5124 | 8.8364 | 0.5532 |
| DPO* | 2.2152 | 16.8396 | 0.4621 | 0.2924 | 8.9597 | 0.5614 |
| NPO* | 1.2611 | 19.6637 | 0.4644 | 0.5151 | 9.0397 | 0.5609 |
| FLAT (Pearson)* | 1.4089 | 15.5543 | 0.4686 | 0.2265 | 8.9906 | 0.5580 |
| ICUL | 1.0121 | 15.6314 | 0.4762 | 2.5585 | 8.9524 | 0.5617 |
| Output Filtering | 2.9832 | 15.6314 | 0.4762 | 0.5292 | 8.9524 | 0.5617 |
| Prompt | 1.3872 | 15.6314 | 0.4762 | 0.4864 | 8.9524 | 0.5617 |
| **GUARD** | 0.6314 | 15.6314 | 0.4762 | 0.1367 | 8.9524 | 0.5617 |

whether the model still retains factual knowledge from the forget set and retain set. ***PrivLeak*** assesses privacy leakage via membership inference (MIA). For detailed definitions and computation procedures, please refer to Appendix F.2.

**GUARD maintains an effective trade-off.** As shown in Table 2, **GUARD** achieves favorable results across multiple evaluation metrics. In terms of *VerbMem* and *KnowMem* on $D_f$, our method significantly reduces memorization risk, with scores of 4.3 and 4.9 respectively, both well below the retained LLM baseline, thus satisfying the unlearning criteria. Furthermore, our method maintains strong performance on *KnowMem* on $D_r$, scoring 55.2, which matches the performance of the original LLM and exceeds all other unlearning baselines except Prompt. These results demonstrate that **GUARD** is effective in removing targeted information while preserving useful knowledge.

## 5.4 COPYRIGHTED CONTENT UNLEARNING

**Experiment setup.** Following prior work (Wang et al., 2024; Liu et al., 2024; Yao et al., 2024b), we use Harry Potter and the Sorcerer's Stone (Rowling, 2023; Eldan & Russinovich, 2023) as the source of copyrighted content to be unlearned. We extract 400 chunks (up to 512 tokens each) from the book to construct the forget set $\mathcal{D}_f$ (Wang et al., 2024; Jia et al., 2024), and sample 400 paragraphs from the C4 dataset (Raffel et al., 2020) to form the retain set $\mathcal{D}_r$. The IDK dataset is taken from (Jia et al., 2024). Following (Wang et al., 2024), we fine-tune OPT-2.7B (Zhang et al., 2022) and Llama2-7B (Touvron et al., 2023) on $\mathcal{D}_f$ to simulate memorization, while the original pre-trained models serve as retained baselines. The objective is to prevent the unlearned model from reproducing copyrighted content.

**Evaluation metrics.** Following the evaluation metrics presented in (Wang et al., 2024), we assess both unlearning effectiveness and model utility. Forgetting is measured using the **Forget Quality Gap (FQ Gap)**, which combines BLEU (Papineni et al., 2002) and ROUGE-L (Lin, 2004) score differences between the unlearned and retained model. Model utility is evaluated via **average accuracy** on nine standard zero-shot benchmarks (Ji et al., 2024), and **perplexity (PPL)** on Wikitext (Merity et al., 2016). Full metric definitions are provided in Appendix F.3.

**Overall, GUARD achieves effective unlearning without compromising model utility.** GUARD achieves the lowest FQ Gap on both OPT-2.7B and Llama2-7B, significantly outperforming all baseline methods. This indicates that its behavior closely aligns with the retained model on forget-specific content, successfully eliminating memorized copyrighted information. In contrast, methods such as GA and KL yield relatively high FQ Gap values, with GA even resulting in an unacceptably large PPL, highlighting a clear trade-off between forgetting and language fluency. Moreover, due to **GUARD** 's training-free nature, it preserves both PPL and average accuracy on nine zero-shot benchmark tasks at levels consistent with the original model across both architectures. While many unlearning methods suffer from a trade-off between improving one metric at the cost of another (e.g.,

lowering PPL while sacrificing accuracy), our method demonstrates superior balance, effectively removing targeted knowledge while maintaining the model's general language understanding and generation capabilities.

Table 4: Impact of different forbidden token methods on **GUARD**, evaluated on the TOFU 1% dataset. Due to the consistency of MU and R-RL with the retain model, we report only FQ and F-RL. The top two metrics are highlighted in blue .

| Methods | FQ($\uparrow$) | F-RL($\downarrow$) |
|---------|------|--------|
| Retained Model | 1.0 | 0.4080 |
| ChatGPT-4o-mini | **0.1649** | **0.3910** |
| Llama2-7B | **0.1649** | **0.4051** |
| All words | **0.1649** | 0.0176 |
| Half words | **0.1649** | 0.0719 |
| Confidence-based | **0.0970** | 0.2160 |

Table 5: Ablation study of **GUARD** 's components, evaluated on the TOFU 1% dataset. We report only FQ and F-RL. The top two metrics are highlighted in blue .

| Methods | FQ($\uparrow$) | F-RL($\downarrow$) |
|---------|------|--------|
| Retained Model | 1.0 | 0.4080 |
| **GUARD** | **0.1649** | **0.3910** |
| w/o Trie | **0.0541** | **0.4243** |
| w/o SBERT | 0.0030 | 0.4967 |

## 5.5 ABLATION STUDIES

**Impact of Forbidden Token Methods on GUARD.** Since **GUARD** requires the extraction of forbidden token from the original answers, different extraction strategies may influence the forget quality. We conducted ablation experiments on the TOFU 1% dataset using the Llama2-7B, comparing the following four forbidden token construction strategies: 1) **Llama2**: using Llama2-7B to replace the ChatGPT-4o-mini (Achiam et al., 2023) in the original method for extraction; 2) **All words**: using all words in the original answer as forbidden token; 3) **Half words**: using only the first half of the words in the original answer; 4) **Confidence-based**: based on the token probabilities generated by the language model, selecting high-confidence content words as forbidden token.

**GUARD maintains strong performance without external models.** Table 4 shows that overall, the FQ performance of these four methods is close to that of the extraction-based approach using ChatGPT-4o-mini, and all significantly outperform the fine-tuned baseline in terms of FQ. However, due to the lack of fine-grained extraction of forbidden token, these methods result in relatively uncontrollable outputs, leading to a deviation in F-RL compared to the retained model. Overall, **GUARD** is able to maintain strong forget quality even without relying on external models.

**Ablation Study of GUARD's Components. Both hard and soft matching are crucial for effective unlearning.** We performed an ablation study to assess the significance of token matching and SBERT-based soft matching, as shown in Table 5. Each module was evaluated individually to verify its effect. The study was conducted using Llama2-7B on the TOFU 1% dataset. Results show that removing any module leads to a decrease in FQ compared to **GUARD**. For F-RL, the absence of either module results in incomplete forgetting, leading to smaller absolute values compared to the retained model. Overall, the combination of token-level hard matching and SBERT-based soft matching improves the generality of unlearning.

## 6 CONCLUSION

In this paper, we introduce **GUARD** (**G**eneration-time **U**nlearning via **A**daptive **R**estriction and **D**etection), a training-free method for LLM unlearning. **GUARD** firstly employs a simple MLP to classify prompts and determine whether they belong to the target categories. It then extracts forbidden token from the original answers and enforces unlearning during generation through a combination of token matching and semantic matching. Extensive experiment results on the TOFU, MUSE, and Harry Potter datasets, as well as the ablation studies, demonstrate that **GUARD** not only significantly outperforms baseline methods in terms of forget quality but also preserves model utility effectively.

ETHICS STATEMENT

We use only publicly available datasets under their respective licenses to evaluate our inference-time unlearning method. Our approach does not retrain or modify LLM parameters, ensuring no additional sensitive information is introduced.

REPRODUCIBILITY STATEMENT

Experimental settings are provided in Appendix E, and the code will be released upon acceptance to support transparency and reproducibility.

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

# APPENDIX

## A    THE USE OF LARGE LANGUAGE MODELS

We used ChatGPT-5 as a writing assistant to improve the clarity and fluency of our manuscript. The model was employed solely for language polishing, including grammar correction, style refinement, and consistency checking. All research ideas, methodological designs, and experimental analyses were fully developed and executed by the authors; the LLM was not involved in generating or interpreting any scientific content.

## B    PROMPT CLASSIFIERS

This section details the training process of the prompt classifiers, including dataset construction and the corresponding evaluation results. We train separate prompt classifiers for three tasks: TOFU (Maini et al., 2024a), HP Book (Wang et al., 2024), and MUSE-News (Shi et al., 2024), aiming to identify inputs that correspond to forget targets. Each classifier is trained as a binary classifier with supervised labels. The data statistics can be found in Table 6.

Table 6: The dataset statistics used to train the prompt classifiers are as follows. Let $D_{\mathrm{P}}^{\mathrm{Train}}$ and $D_{\mathrm{N}}^{\mathrm{Train}}$ represent the positive and negative training sets, respectively. The test set consists of $D^{\mathrm{Test}}$, $D_{P_{\mathrm{para}}}^{\mathrm{Test}}$, and $D_{N_{\mathrm{para}}}^{\mathrm{Test}}$, where $D^{\mathrm{Test}}$ is the combination of the TOFU dataset's real authors and world facts sets. The other two subsets are composed of paraphrased versions of the positive and negative samples, respectively. Additionally, $D_{\mathrm{g}}^{\mathrm{Test}}$ refers to the general test set, which is used to evaluate the model's overall utility. The dataset also includes two tasks from the MUSE-News collection: News (*knowmem*), focusing on memory retention of factual knowledge, and News (*verbmem*), assessing memory retention on a per-line basis.

| Dataset | $D_{\mathrm{P}}^{\mathrm{Train}}$ | $D_{\mathrm{N}}^{\mathrm{Train}}$ | $D^{\mathrm{Test}}$ | $D_{P_{\mathrm{para}}}^{\mathrm{Test}}$ | $D_{N_{\mathrm{para}}}^{\mathrm{Test}}$ | $D_{\mathrm{g}}^{\mathrm{Test}}$ |
|---|---|---|---|---|---|---|
| TOFU (1%) | 880 | 86,449 | 217 | 160 | 15,840 | 29,590 |
| TOFU (5%) | 4,200 | 86,888 | 217 | 800 | 15,200 | 29,590 |
| TOFU (10%) | 8,800 | 82,488 | 217 | 1,600 | 14,400 | 29,590 |
| HP Book | 353,470 | 346,963 | - | 141,388 | 137,470 | 29,590 |
| News (*knowmem*) | 2,200 | 5,488 | - | 400 | 400 | 29,590 |
| News (*verbmem*) | 900 | 12,288 | - | 200 | 2,000 | 29,590 |

## B.1 TRAINING DATASETS

**TOFU dataset.** We follow the original data splits provided by the TOFU dataset (Maini et al., 2024a). Specifically, TOFU defines forget sets at 1%, 5%, and 10%, which we use as positive samples, with the corresponding retain data serving as negative samples.

**HP book.** To prevent models from revealing copyrighted content, we train a prompt classifier targeting passages from Harry Potter and the Sorcerer's Stone (Rowling, 2023). Positive samples are extracted from the official eBook using spaCy's `sentencizer`, and we retain only sentences longer than 20 characters to avoid structural or low-content artifacts. Negative samples are drawn from the BookMIA dataset (Shi et al., 2023), with all Harry Potter-related content removed. Since generalization is not the focus of this task, no additional test set is introduced.

**MUSE-News.** Since the MUSE-News (Shi et al., 2024) includes two tasks, including *knowmem* and *verbmem*, we trained two separate classifiers for these tasks. For *knowmem*, we used forget data and retain data as positive and negative samples, respectively. Since *knowmem* mainly tests the model's ability to retain information from QA pairs, we constructed modified prompts, adversarial prompts, irrelevant context prompts, and jailbreak prompts, similar to the approach used in TOFU. On the other hand, *verbmem* focuses on testing the model's ability to retain memory on a per-line basis. For this task, we used forget data as the positive samples. For negative samples, we used the CC News dataset (Hamborg et al., 2017) and randomly sampled 1,000 data points for this purpose.

**General utility evaluation.** In real-world applications, it is important not only to distinguish retain/forget targets, but also to preserve the model's ability to recognize general tasks. To this end, we introduce an auxiliary evaluation set that includes four commonly used LLM benchmarks: BoolQ (Clark et al., 2019), RACE (Lai et al., 2017), SQuAD (Rajpurkar et al., 2016), and TriviaQA (Joshi et al., 2017). Together, they contain 32,877 samples. We use 10% of this data for training and the remaining 90% for testing, allowing us to measure the classifier's behavior on o.o.d. and utility-preserving prompts.

## B.2 PROMPT VARIANTS FOR ROBUSTNESS EVALUATION

Although existing benchmarks (e.g., TOFU) do not require measuring generalization ability, real-world scenarios often involve noisy, paraphrased, or even adversarial user inputs. In practice, users may attempt to bypass forget classifiers using sophisticated rewriting techniques or jailbreak attacks. To better simulate these deployment challenges, we introduce several types of perturbed prompts to evaluate classifier robustness.

Specifically, for each benchmark dataset, we augment the original forget and retain prompts with the following variations:

- **Paraphrased Prompts:** Surface-form rewrites that preserve the original semantics, generated via ChatGPT-4o-mini. These mimic natural rewording by users.

- **Adversarial Prompts:** Intentionally engineered inputs that preserve the source semantics under perturbation while introducing lexical, syntactic, character-level and encoding/formatting alterations. These prompts are constructed to lie near the classifier's decision boundary and induce misclassification with high semantic fidelity.

- **Jailbreak Prompts:** We incorporate several commonly observed jailbreak patterns from real-world LLM usage (Yi et al., 2024). These prompts are crafted by adding special prefixes or suffixes that aim to manipulate the prompt context without modifying its core intent, thereby attempting to evade detection through indirect phrasing.

- **Irrelevant Context Prompts:** Extraneous unrelated text is prepended to the original prompt to introduce distractive noise. The length of the added distractive context ranges from 50 to 500 textual units (including words and punctuation).

These perturbed prompts are used during both training and evaluation to assess classifier robustness under distribution shift and adversarial threat models.

Table 7: The false negative rate (FNR) and false positive rate (FPR) of the prompt classifiers on various datasets are as follows. $D_{\text{ori}}^{\text{Train}}$ represents the test results of the original prompts on each benchmark, while $D_{\text{rephara}}^{\text{Test}}$, $D_{\text{adv}}^{\text{Test}}$, $D_{\text{irr}}^{\text{Test}}$, and $D_{\text{jail}}^{\text{Test}}$ represent the results on the paraphrased prompt test set, the adversaria prompt test set and the jailbreak attack prompt test set. The $D_{\text{g}}^{\text{Test}}$ set contains out-of-distribution prompts from four benchmarks.

(a) The FNR of each dataset.

| Dataset | $\text{FNR}_{D_{\text{ori}}^{\text{Train}}}$ | $\text{FNR}_{D_{\text{rephara}}^{\text{Test}}}$ | $\text{FNR}_{D_{\text{adv}}^{\text{Test}}}$ | $\text{FNR}_{D_{\text{irr}}^{\text{Test}}}$ | $\text{FNR}_{D_{\text{jail}}^{\text{Test}}}$ |
|---|---|---|---|---|---|
| TOFU (1%) | 0.0 | 0.0256 | 0.0256 | 0.0256 | 0.0 |
| TOFU (5%) | 0.0 | 0.0015 | 0.0065 | 0.0400 | 0.0025 |
| TOFU (10%) | 0.0 | 0.0100 | 0.0429 | 0.0175 | 0.0049 |
| HP Book | 0.0 | - | - | 0.0 | 0.0 |
| News (*knowmem*) | 0.0 | 0.0100 | 0.0208 | 0.0392 | 0.0099 |
| News (*verbmem*) | 0.0 | - | - | 0.0 | 0.0 |

(b) The FPR of each dataset.

| Dataset | $\text{FPR}_{D_{\text{ori}}^{\text{Train}}}$ | $\text{FPR}_{D^{\text{Test}}}$ | $\text{FPR}_{D_{\text{rephara}}^{\text{Test}}}$ | $\text{FPR}_{D_{\text{adv}}^{\text{Test}}}$ | $\text{FPR}_{D_{\text{irr}}^{\text{Test}}}$ | $\text{FPR}_{D_{\text{jail}}^{\text{Test}}}$ | $\text{FPR}_{D_{\text{g}}^{\text{Test}}}$ |
|---|---|---|---|---|---|---|---|
| TOFU (1%) | 0.0 | 0.0 | 0.0002 | 0.0 | 0.0 | 0.0002 | 0.0004 |
| TOFU (5%) | 0.0 | 0.0 | 0.0003 | 0.0008 | 0.0047 | 0.0003 | 0.0021 |
| TOFU (10%) | 0.0 | 0.0 | 0.0011 | 0.0011 | 0.0013 | 0.0008 | 0.0033 |
| HP Book | 0.0 | - | - | - | 0.0004 | 0.0002 | 0.0057 |
| News (*knowmem*) | 0.0 | - | 0.0 | 0.0 | 0.0 | 0.0100 | 0.0056 |
| News (*verbmem*) | 0.0 | - | - | - | 0.0 | 0.0 | 0.0001 |

## B.3 TRAINING PROCESS

For all classifiers, we use a simple MLP for training. The structure of the MLP includes an input layer, a hidden layer, and an output layer. The hidden layer uses the ReLU (Nair & Hinton, 2010) activation function, with Dropout and LayerNorm applied to prevent overfitting and accelerate convergence. The final output layer uses a linear transformation to produce classification results. The input to the model is the average of the penultimate layer embeddings from the LLM for each prompt. The advantage of this approach is that it eliminates the need for additional models, relying solely on a simple MLP for classification. Here, we use OPT-2.7B (Zhang et al., 2022) for extracting embeddings. Since, in most cases, the number of positive samples (forget samples) is much smaller than the negative samples, we re-weight the class-level loss using inverse frequency.

## B.4 EXPERIMENTAL RESULTS

Table 7 summarizes the performance of our prompt classifiers across different benchmarks and attack settings. We observe that despite its simplicity, our MLP-based classifier demonstrates strong robustness and generalization in five key aspects as detailed below.

**Strong performance on original and paraphrased prompts.** Table 7 shows that the simple MLP classifier attains $0\%$ error on in-domain (original) prompts across all benchmarks, i.e., both $\text{FNR}_{D_{\text{ori}}^{\text{Train}}}$ and $\text{FPR}_{D_{\text{ori}}^{\text{Train}}}$ are 0. On paraphrased test sets ($D_{\text{rephara}}^{\text{Test}}$), the classifier continues to perform well: FPR remains near zero on all datasets, while FNR stays at low single-digit percentages depending on the split (e.g., TOFU 1%/5%/10%). These results indicate that the MLP learned decision boundaries that are stable to surface-form rewrites without sacrificing precision.

**Robustness to class imbalance.** Despite pronounced class skew in several settings, the classifier remains stable and does not collapse toward the majority class. For example, the TOFU splits exhibit highly imbalanced positive-to-negative ratios, yet the in-domain evaluation on original prompts achieves $0\%$ error (both $\text{FNR}_{D_{\text{ori}}^{\text{Train}}}$ and $\text{FPR}_{D_{\text{ori}}^{\text{Train}}}$ are 0; see Tables 6 and 7). Similarly, other benchmarks with sizable or skewed base rates (e.g., HP Book and News) also show $0\%$ error on originals. These results indicate that the learned decision rule preserves sensitivity to positives and specificity to negatives even under severe prior imbalance, suggesting robust calibration with respect to class prevalence.

Table 8: Retrieval accuracy of similarity search across different benchmarks.

| Dataset | SBERT | | SBERT+RoBerta | |
|---|---|---|---|---|
| | Acc. | Time (ms) | Acc. | Time (ms) |
| TOFU 1% | 0.9463 | 0.10 | 0.9744 | 5.61 |
| TOFU 5% | 0.9186 | 0.09 | 0.9724 | 5.59 |
| TOFU 10% | 0.9070 | 0.10 | 0.9637 | 5.71 |
| MUSE-News | 1.0 | 0.09 | 1.0 | 8.41 |

**Robustness to adversarial and jailbreak attacks.** Under adversarial perturbations ($D_{\mathrm{adv}}^{\mathrm{Test}}$) and jailbreak prompts ($D_{\mathrm{jail}}^{\mathrm{Test}}$), the classifier maintains low error rates. For jailbreak specifically, FNR is near zero across datasets (e.g., TOFU and HP book are at or close to 0; News *knowmem* is below 1%), and FPR is also minimal (at most around 1% on *knowmem*, and near zero elsewhere). Adversarial prompts are more challenging, leading to slightly higher FNR on certain TOFU splits (still within low single digits), while FPR remains very small. Overall, these results suggest resilience to both boundary-level edits and prefix/suffix-based evasion attempts.

**Stability under long/noisy contexts.** When unrelated context is prepended ($D_{\mathrm{irr}}^{\mathrm{Test}}$), the classifier remains stable: FPR stays low across datasets (e.g., near zero for TOFU 1%/10% and News *verbmem*, and $< 0.5\%$ for HP book), and FNR increases only moderately on the most difficult TOFU 5% case, while remaining low elsewhere. Moreover, the MUSE-News and HP book settings—both involving longer passages—show consistently low error rates, indicating that the MLP is robust to longer inputs and distractive noise.

**Good performance on out-of-distribution general prompts.** On the out-of-distribution general set ($D_{\mathrm{g}}^{\mathrm{Test}}$) built from four standard LLM benchmarks, the classifier exhibits uniformly low FPR (well below 1% across all datasets). This suggests that the prompt classifier does not overfire on unrelated utility prompts and therefore is unlikely to harm general model behavior outside the forgetting scope.

## C  SIMILARITY RETRIEVAL

When a sample is classified as belonging to the forget target, we retrieve the original answer from the forget data to facilitate subsequent forbidden token extraction. Since intra-domain matching effectively involves retrieving each prompt against itself, it trivially achieves 100% accuracy. Therefore, we focus exclusively on evaluating the retrieval top-1 accuracy between rewritten prompts and their original counterparts. Furthermore, we do not include tasks such as the HP Book and MUSE-News *verbmem*, as these primarily evaluate a model's ability to continue passages based on original book or news excerpts, where the prompts must contain content almost identical to the original text. Therefore, in this study, we restrict our focus to QA pair-based matching, specifically for the TOFU dataset and the *knowmem* task in MUSE-News.

We adopt a simple SBERT-based[2] similarity retrieval approach. Specifically, for each rewritten prompt, we perform pairwise matching and evaluate the top-1 retrieval accuracy. Table 8 summarizes our experimental results. Without any task-specific fine-tuning, but using only the pretrained model weights, we observe that the retrieval top-1 accuracy reaches above 90%. Since our main focus here is on exploring zero-shot performance, we further enhance the matching process by first retrieving the top-5 candidates using SBERT, followed by a second-stage reranking using the Roberta[3] model. This two-stage process improves the retrieval top-1 accuracy by an additional 5% on average. We also report the average inference time for matching. Our results suggest that even without fine-tuning, existing pretrained similarity models can achieve high efficiency and accuracy, and that further fine-tuning could potentially lead to even better performance.

## D  BASELINE METHODS

In this section, we introduce the baseline methods used in our paper.

---

[2]sentence-transformers/paraphrase-MiniLM-L6-v2

[3]cross-encoder/stsb-roberta-base

**In-Context Unlearning (ICUL) (Pawelczyk et al., 2023).** ICUL is a training-free method that removes the influence of specific data points from a language model by manipulating the in-context examples during inference, without updating the model parameters. To unlearn a target point, ICUL constructs a prompt that includes the point with a randomly flipped label (or incorrect answer) and augments it with several correctly labeled examples drawn from the training distribution. This design aims to diminish the model's confidence on the forgotten points, making its behavior resemble that of a retrained model excluding those points. The constructed prompt follows the format:

---

**The Prompt Used in ICUL**

[Forget Input 1] [Different Label]    . . .    [Forget Input K] [Different Label] [Correct Input 1] [Correct Label 1]    . . .    [Correct Input L] [Correct Label L]  [Query Input]

---

Inference is performed using this prompt with deterministic decoding (temperature t = 0), effectively simulating the model's output as if the forget points had never been seen during training.

**Output Filtering (Thaker et al., 2024).** Output filtering is a lightweight, training-free strategy that aims to suppress model outputs containing forgotten information without modifying model parameters. In this method, after the model generates a candidate response, a filter model or rule-based system is applied to post-process the output. If the output is detected to contain sensitive or forgotten content, the response is not returned as-is; instead, it is replaced with a fixed template answer: *"I'm not sure"*. To determine whether a response contains sensitive information, simple classifiers, keyword-based matching, or large models (such as GPT-4) can be used. For simplicity, this paper assumes an idealized setting where all sensitive outputs are perfectly detected without false positives or false negatives.

**Prompt Baseline.** Inspired by the prompt-based unlearning strategies proposed in Pawelczyk et al. (2023); Liu et al. (2024); Muresanu et al. (2024); Bhaila et al. (2024), we implement a simple prefix-tuning baseline. In this approach, the model is guided to suppress memorized or undesired responses by prepending a system-level instruction that explicitly discourages content disclosure. The prompt used in our experiments is as follows:

---

**The Prompt Used in Prompt Baseline**

Instruction: Please note: As the user's question involves sensitive content, your response should either avoid providing related knowledge or explicitly state that such information cannot be provided. Additionally, try to avoid repeating previous responses—offer a different perspective if possible, or indicate that there is insufficient information available.
User question: {question}
Please respond accordingly.

---

**Gradient Ascent (GA) (Yao et al., 2024b).** Gradient ascent is an optimization technique that adjusts model parameters in the direction that increases a given objective function. In unlearning scenarios, GA is often applied to intentionally increase the prediction loss over the forget dataset $D_f$, thus encouraging the model to move away from representations learned from $D_f$. This process implicitly counteracts prior learning on the forget data, guiding the model toward a state that resembles training on the retain set $D_r$ alone. The corresponding loss function can be formulated as:

$$\mathcal{L}_{\text{GA}} = -\frac{1}{|D_f|} \sum_{i=1}^{|D_f|} \ell(x_i, y_i; \theta). \tag{12}$$

**GradDiff (GD) (Liu et al., 2024).** Gradient Difference is an optimization-based unlearning strategy that jointly applies opposing gradient signals over two disjoint datasets. Specifically, it encourages the model to degrade its performance on the forget set $\mathcal{D}_f$ via loss maximization, while simultaneously preserving its behavior on the retain set $\mathcal{D}_r$ through conventional minimization. This dual objective can be captured by the following composite loss:

$$\mathcal{L}_{\text{GD}} = -\mathcal{L}(\mathcal{D}_f; \theta) + \mathcal{L}(\mathcal{D}_r; \theta). \tag{13}$$

**KL Minimization (KL) (Maini et al., 2024a).** This method encourages the model to forget unwanted information while maintaining alignment with its original behavior on retained data. Specifically, it penalizes deviations from the original model's output distribution on the retain set $\mathcal{D}_r$ using Kullback–Leibler (KL) divergence, while simultaneously promoting forgetting by increasing the loss on the forget set $\mathcal{D}_f$. Let $\mathcal{M}_\theta$ denote the current model, and $\mathcal{M}_{\hat{\theta}}$ the original (pre-unlearning) model. The combined objective can be written as:

$$\mathcal{L}_{\text{KL}} = -\mathcal{L}(\mathcal{D}_f; \theta) + \frac{1}{|\mathcal{D}_r|} \sum_{x \in \mathcal{D}_r} \frac{1}{|x|} \sum_{i=2}^{|x|} \text{KL}\left(\mathcal{M}_\theta(x_{\leq i}) \parallel \mathcal{M}_{\hat{\theta}}(x_{\leq i})\right). \tag{14}$$

**Preference optimization (PO) (Maini et al., 2024a).** This approach enforces unlearning by modifying the model's response preferences. Instead of generating factual or detailed answers for samples in the forget set $\mathcal{D}_f$, the model is trained to produce safe refusal responses such as "I'm unable to answer that". This transformation yields a derived dataset $\mathcal{D}_{\text{IDK}}$, which pairs the original queries with target refusal completions. To simultaneously retain the model's performance on trusted data, training minimizes the following objective:

$$\mathcal{L}_{\text{PO}} = \mathcal{L}(\mathcal{D}_{\text{IDK}}; \theta) + \mathcal{L}(\mathcal{D}_r; \theta). \tag{15}$$

**Direct Preference Optimization (DPO) (Rafailov et al., 2023).** To remove specific knowledge while preserving overall model behavior, this approach adapts the Direct Preference Optimization (DPO) framework to the unlearning context. Instead of contrasting human-preferred and less-preferred responses, the loss compares a target refusal output $y_e$ with the original (to-be-forgotten) response $y_f$ under the same input $x_f \in \mathcal{D}_f$. Let $\beta$ be the inverse temperature, the unlearning objective is defined as:

$$\mathcal{L}_{\text{DPO}} = -\frac{2}{\beta} \mathbb{E}_{\mathcal{D}_f} \left[ \log \sigma \left( \beta \log \prod_{i=1}^{|y_e|} h_\theta(x_f, y_{e, <i}) - \beta \log \prod_{i=1}^{|y_f|} h_\theta(x_f, y_{f, <i}) - \mathcal{M}_{\text{ref}} \right) \right]. \tag{16}$$

Here, $h_\theta(\cdot)$ denotes the model's next-token predictive distribution, and $\mathcal{M}_{\text{ref}}$ optionally penalizes deviation from the original model to preserve retention. The DPO loss encourages the model to prefer safe completions $y_e$ over original responses $y_f$, thus enforcing targeted forgetting.

To better preserve model utility while performing targeted forgetting, we further introduce the retention-regularized variant of DPO:

$$\mathcal{L}_{\text{DPO-RT}} = \mathcal{L}_{\text{DPO}} + \mathcal{L}_{\text{r}}, \tag{17}$$

where $\mathcal{L}_{\text{r}}$ denotes the supervised loss on the retain set $\mathcal{D}_r$, encouraging the model to maintain desirable knowledge while forgetting specific content.

**Negative Preference Optimization (NPO) (Zhang et al., 2024).** The NPO method focuses on suppressing undesired responses by penalizing the likelihood of preferred completions within the forget set $\mathcal{D}_f$. Unlike Direct Preference Optimization (DPO), which contrasts preferred and dispreferred responses, NPO only utilizes the dispreferred term, aiming for more targeted unlearning. Let $\beta$ be the inverse temperature scaling factor and $|\mathcal{D}_f|$ the size of the forget set, the NPO objective is defined as:

$$\mathcal{L}_{\text{NPO}} = \frac{2}{\beta |\mathcal{D}_f|} \sum_{(x,y) \in \mathcal{D}_f} \log \left( 1 + \left( \frac{h_\theta(y \mid x)}{h_\theta(y \mid x)} \right)^\beta \right). \tag{18}$$

To ensure utility preservation, we consider the retention-regularized variant of NPO, which incorporates supervised fine-tuning on the retain set $\mathcal{D}_r$:

$$\mathcal{L}_{\text{NPO-RT}} = \mathcal{L}_{\text{NPO}} + \mathcal{L}_{\text{r}}. \tag{19}$$

**Mismatch.** Mismatch retains the same objective as the preference-optimization framework described above, but additionally constructs a random combination of text sequences $\mathbf{x}_{\text{rand}}$. In this formulation, the second term of the Mismatch loss is identical to the second term in LLMU (Yao et al., 2024b):

$$\mathcal{L}_{\text{Mismatch}} = \mathcal{L}_{\text{Fine-tune}} \; + \; \frac{1}{|D_{\text{rand}}|} \sum_{x \in D_{\text{rand}}} \mathcal{L}(x; \theta). \tag{20}$$

**LLMU (Yao et al., 2024b).** LLMU combines the GA term with two auxiliary components: (1) random-completion unlearning on $\mathcal{D}_{\text{rand}}$ (constructed from prompts in $\mathcal{D}_f$) and (2) retention regularization on $\mathcal{D}_{\text{normal}}$. In our setup we fix $\epsilon_2 = \epsilon_3 = 1$ and tune $\epsilon_1 \in \{0.1, 0.5, 1, 2\}$.

$$\begin{aligned} \mathcal{L}_{\text{LLMU}} = &- \frac{\epsilon_1}{|\mathcal{D}_f|} \sum_{x \in \mathcal{D}_f} \mathcal{L}(x; \theta) \; + \; \frac{\epsilon_2}{|\mathcal{D}_{\text{rand}}|} \sum_{x \in \mathcal{D}_{\text{rand}}} \mathcal{L}(x; \theta) \\ &+ \frac{\epsilon_3}{|\mathcal{D}_{\text{normal}}|} \sum_{x \in \mathcal{D}_{\text{normal}}} \text{KL}\big(h(x; \theta_o) \,\|\, h(x; \theta)\big). \end{aligned} \tag{21}$$

**Task Vectors (Eldan & Russinovich, 2023).** The task vector method constructs an unlearned model by explicitly subtracting the direction of adaptation on the forget set $\mathcal{D}_f$. Let $\theta_o$ denote the parameters of the original language model, and $\theta_{\text{reinforce}}$ be the model fine-tuned to overfit $\mathcal{D}_f$. Then, the unlearned model $\theta$ is computed by reversing the adaptation vector:

$$\theta = \theta_o - (\theta_{\text{reinforce}} - \theta_o). \tag{22}$$

This effectively moves the model away from the representation learned from $\mathcal{D}_f$, without additional optimization.

**Who's Harry Potter (WHP) (Eldan & Russinovich, 2023).** WHP defines the unlearned model in terms of a distributional interpolation between the original model $\theta_o$ and the reinforced model $\theta_{\text{reinforce}}$. Let $p_\theta(\cdot \mid x)$ denote the token-level output distribution for a given input $x$. WHP then adjusts the generation probabilities as:

$$p_\theta(\cdot \mid x) = p_{\theta_o}(\cdot \mid x) - \alpha \left( p_{\theta_{\text{reinforce}}}(\cdot \mid x) - p_{\theta_o}(\cdot \mid x) \right), \tag{23}$$

where $\alpha$ is a tunable coefficient that governs the extent of unlearning by controlling how far the resulting distribution is pushed away from $p_{\theta_{\text{reinforce}}}$.

**FLAT (Wang et al., 2024).** Forget data only Loss AjustmenT (FLAT) is a loss adjustment-based unlearning method that eliminates the need for retain data or a reference model. Instead of performing direct gradient ascent on forget data, FLAT leverages f-divergence maximization between a preferred template response and the original forget response to guide unlearning. For each forget sample $(x_f, y_f)$, a manually designed or generated template response $y_e$ (such as a refusal or irrelevant answer) is paired. FLAT optimizes a composite loss that encourages the model to move closer to $y_e$ while forgetting $y_f$, formulated as:

$$\mathcal{L}_{\text{FLAT}} = -g^* \left( P(x_f, y_e; \theta) \right) + f^* \left( g^* \left( P(x_f, y_f; \theta) \right) \right), \tag{24}$$

where $P(x_f, y; \theta)$ denotes the average token prediction probability for response $y$ given prompt $x_f$, $g^*(\cdot)$ and $f^*(\cdot)$ are the optimal variational and conjugate functions corresponding to a chosen f-divergence. This formulation allows FLAT to assign appropriate importance to learning from template responses and forgetting undesired ones, achieving strong unlearning performance without sacrificing overall model utility.

# E  EXPERIMENT SETUP

## E.1  BASELINE SETUP

We conduct fine-tuning for all original models under consistent hyperparameter settings to ensure comparability. For the TOFU dataset, we adopt a batch size of 32, aligning with previous studies (Wang et al., 2024; Maini et al., 2024a; Zhang et al., 2024; Ji et al., 2024). Both OPT-2.7B and Phi-1.5B models are fine-tuned from their pretrained checkpoints for 5 epochs using a learning rate of $2 \times 10^{-5}$. LLaMA2-7B is similarly fine-tuned for 5 epochs but with a lower learning rate of $1 \times 10^{-5}$. All fine-tuning procedures employ the AdamW (Loshchilov & Hutter, 2017) optimizer. During the unlearning phase, we retain the same learning rate configurations used in the original fine-tuning stage to maintain consistency.

For the HP Book dataset, we adopt the hyperparameter settings reported in (Wang et al., 2024) to train the original model. Additionally, for MUSE-News, we utilize the official pretrained models released by the original authors[4] to conduct our experiments.

## E.2  **GUARD** SETUP

In our method, it is necessary to extract forbidden token from the original answers to facilitate subsequent unlearning operations. Different extraction strategies are adopted depending on the application scenario. For the TOFU dataset, the metrics reported in Sec.5.2 are based on forbidden token extracted using ChatGPT-4o-mini. This approach enables more effective identification of key phrases within the original answers, thereby allowing **GUARD** to perform more precise unlearning. However, it is important to note that the use of ChatGPT-4o-mini serves solely to establish the theoretical upper bound of unlearning performance. We also report results in Sec.5.5 using alternative extraction strategies, including methods that do not require the introduction of external models. The experiments demonstrate that **GUARD** can still achieve strong forget quality without relying on additional models for forbidden token extraction.

For the MUSE-News datasets, since the primary objective is to prevent the model from exactly reproducing the original content, we directly use either all words from the original answers or the first half of the words as the forbidden token for processing. We use 2 H20 GPUs to run all experiments.

Additionally, since **GUARD** relies on beam search, token-level hard matching, and SBERT-based soft matching to implement generation-time unlearning, we adopt a beam width of 7, set the hard matching threshold $\beta$ to 1, and fix the similarity threshold $\delta$ for soft matching to 0.5 in all experiments. We provide a detailed discussion on the impact of different hyperparameter settings in Appendix H.

# F  EVALUATION METRICS

## F.1  TOFU

**Probability.** For each instance in either the retain or forget set, we compute the normalized conditional probability $P(a \mid q)^{1/|a|}$, where $q$ denotes the input question, $a$ represents the answer, and $|a|$ is the number of tokens in $a$. In the real authors and world facts subsets, the dataset provides five candidate answers $\{a_0, \tilde{a}_1, \tilde{a}_2, \tilde{a}_3, \tilde{a}_4\}$, where $a_0$ is the correct answer and the $\tilde{a}_i$ are perturbed (incorrect) alternatives. The probability ratio is calculated as:

$$\text{Probability} = \frac{P(a_0 \mid q)^{1/|a_0|}}{\sum_{i=1}^{4} P(\tilde{a}_i \mid q)^{1/|\tilde{a}_i|}}. \tag{25}$$

**Truth Ratio.** The truth ratio measures the model's preference for perturbed answers. It is computed as the geometric mean of the normalized probabilities of all perturbed answers $\{\tilde{a}_1, \tilde{a}_2, \dots\}$ relative

---

[4]muse-bench/MUSE-news_target

to the normalized probability of the paraphrased answer $\hat{a}$:

$$R_{\text{truth}} = \frac{\left(\prod_{i=1}^{|\mathcal{A}|} P(\tilde{a}_i \mid q)^{1/|\tilde{a}_i|}\right)^{1/|\mathcal{A}|}}{P(\hat{a} \mid q)^{1/|\hat{a}|}}. \tag{26}$$

In the real authors and world facts subsets, since paraphrased answers are unavailable, the original answer $a$ is used in the denominator.

**ROUGE-L.** For all TOFU subsets, we report the ROUGE-L recall score (Lin, 2004) between the ground truth answers (forget dataset) and the model outputs after unlearning.

**Model Utility.** Model utility is calculated as the harmonic mean of nine scores, covering answer probability, truth ratio, and ROUGE-L recall across the retain, real authors, and world facts subsets. A higher utility score indicates better overall performance.

**Forget Quality.** Forget quality is evaluated by applying a Kolmogorov-Smirnov (KS) test to compare the distributions of truth ratios from the retained and unlearned models on the forget set. A higher $p$-value supports the null hypothesis that the two distributions are identical, indicating similar behavior between the retained and unlearned models.

### F.2 MUSE

**No Verbatim Memorization.** To evaluate whether a model has fully unlearned specific content, we assess verbatim memorization (*VerbMem*). This metric measures the similarity between the model's continuation output and the ground-truth continuation from the forget set, based on the first $l$ tokens of each sample. The ROUGE-L F1 score (Lin, 2004) is used for evaluation:

$$\text{VerbMem}(f, \mathcal{D}) := \frac{1}{|\mathcal{D}_{\text{forget}}|} \sum_{x \in \mathcal{D}_{\text{forget}}} \text{ROUGE}(f(x_{[:l]}), x_{[l+1:]}). \tag{27}$$

**No Knowledge Memorization.** Knowledge memorization (*KnowMem*) assesses whether the model retains information about the forgotten records. For each question-answer pair $(q, a)$ in the forget set $\mathcal{D}_{\text{forget}}$, we compute the ROUGE score between the model's predicted answer $f(q)$ and the ground-truth $a$, and then average across all examples:

$$\text{KnowMem}(f, \mathcal{D}_{\text{forget}}) := \frac{1}{|\mathcal{D}_{\text{forget}}|} \sum_{(q,a) \in \mathcal{D}_{\text{forget}}} \text{ROUGE}(f(q), a). \tag{28}$$

**No Privacy Leakage.** Privacy leakage is evaluated by assessing whether membership information from the forget set can be inferred. This is measured via membership inference attacks (MIA) that leverage loss statistics to distinguish between training examples (members) and non-training examples (non-members). Following (Murakonda et al., 2021; Ye et al., 2022), the privacy leakage metric, PrivLeak, is defined based on the difference in AUC-ROC scores between the unlearned and retrained models:

$$\begin{aligned} \text{PrivLeak} := \frac{\text{AUC}\big(f_{\text{unlearn}}, \mathcal{D}_{\text{forget}}, \mathcal{D}_{\text{holdout}}\big)}{\text{AUC}\big(f_{\text{retrain}}, \mathcal{D}_{\text{forget}}, \mathcal{D}_{\text{holdout}}\big)} \\ - 1. \end{aligned} \tag{29}$$

A well-performing unlearning algorithm is expected to achieve a PrivLeak score close to zero, while significant positive or negative values indicate issues with over-unlearning or under-unlearning, respectively.

**Utility Preservation.** Utility preservation evaluates whether the model retains its general capabilities after unlearning. We measure the model's performance on the retain set $\mathcal{D}_{\text{retain}}$ by computing the knowledge memorization score:

$$\text{KnowMem}(f_{\text{unlearn}}, \mathcal{D}_{\text{retain}}). \tag{30}$$

Table 9: Runtime comparison across decoding strategies with batch size 128. Batch time is the total wall-clock time to produce outputs for 128 prompts. Single-query latency is measured with 5 warm-up runs followed by 30 measured runs per prompt (120 total per mode). All numbers are in milliseconds (ms).

| Decoder | Batch (128 prompts) | Single-query latency (120 runs) | | |
|---|---|---|---|---|
| | Time [ms] ↓ | Mean [ms] ↓ | Median [ms] ↓ | p95 [ms] ↓ |
| Greedy | 1057 | 180.9 | 185.3 | 194.6 |
| Beam (B=7) | 14377 | 383.3 | 377.3 | 596.1 |
| **GUARD** (uncache) | 42681 | 546.1 | 525.6 | 729.4 |
| **GUARD** (cache) | 17735 | 417.7 | 397.0 | 584.7 |

Table 10: Evaluation results on 5% TOFU dataset. Metrics include FQ, MU, R-RL, and F-RL. The top two performing methods are marked with blue.

| Base LLM | Llama2-7B | | | | Phi-1.5B | | | | OPT-2.7B | | | |
|---|---|---|---|---|---|---|---|---|---|---|---|---|
| Metric | FQ(↑) | MU(↑) | F-RL(↓) | R-RL(↑) | FQ(↑) | MU(↑) | F-RL(↓) | R-RL(↑) | FQ(↑) | MU(↑) | F-RL(↓) | R-RL(↑) |
| Original LLM | 3.4320e-16 | 0.6247 | 0.9756 | 0.9819 | 6.5408e-13 | 0.5194 | 0.9321 | 0.9276 | 3.4320e-16 | 0.5111 | 0.8692 | 0.8807 |
| Retained LLM | 1.0 | 0.6005 | 0.3980 | 0.9798 | 1.0 | 0.5249 | 0.4285 | 0.9159 | 1.0 | 0.5002 | 0.3894 | 0.8660 |
| GA | 8.0566e-07 | 0.0 | 0.0038 | 0.0031 | 3.3925e-18 | 0.0 | 0.0002 | 0.0001 | 2.6127e-07 | 0.0 | 0.0 | 0.0 |
| KL | 4.8692e-10 | 0.4550 | 0.0155 | 0.5758 | 8.7540e-18 | 0.0 | 0.0001 | 0.0001 | 2.6127e-07 | 0.0 | 0.0 | 0.0 |
| GD | 2.3797e-06 | 0.0 | 0.0045 | 0.0040 | 1.1150e-05 | 0.3571 | 0.0014 | 0.4525 | 1.3921e-06 | 0.4297 | 0.0297 | 0.4104 |
| LLMU | 2.9607e-05 | 0.0 | 0.0062 | 0.0071 | 3.9210e-07 | 2.0130e-31 | 0.0652 | 0.0671 | 1.8266e-05 | 0.0 | 0.0080 | 0.0076 |
| PO | 1.3921e-06 | 0.0 | 0.0035 | 0.0032 | 4.8692e-10 | 0.4569 | 0.1897 | 0.7052 | 1.3261e-13 | 0.3555 | 0.0377 | 0.6884 |
| DPO-RT | 1.1150e-05 | 0.0 | 0.0177 | 0.0151 | **0.0220** | 0.0356 | 0.1951 | 0.1960 | **0.1122** | 0.0 | 0.0136 | 0.0144 |
| NPO-RT | **0.1779** | 0.2961 | **0.3332** | 0.4015 | **0.0521** | 0.3999 | **0.4269** | 0.4745 | **0.0521** | 0.4182 | **0.2213** | 0.3548 |
| FLAT (Pearson) | 4.3551e-23 | 0.1476 | 0.0175 | 0.1467 | 0.0002 | 0.5023 | 0.2498 | 0.7021 | 3.0799e-12 | 0.5084 | 0.0157 | 0.6306 |
| ICUL | 3.0799e-12 | **0.6247** | 0.5436 | **0.9819** | 4.4486e-08 | **0.5194** | 0.0577 | **0.9276** | 5.9510e-11 | **0.5111** | 0.0868 | **0.8807** |
| Output Filtering | 5.6169e-17 | **0.6247** | 0.0006 | **0.9819** | 3.1330e-21 | **0.5194** | 0.0006 | **0.9276** | 4.9085e-19 | **0.5111** | 0.0006 | **0.8807** |
| Prompt | 1.1087e-14 | **0.6247** | 0.4886 | **0.9819** | 4.8692e-10 | **0.5194** | 0.1042 | **0.9276** | 1.1087e-14 | **0.5111** | 0.7343 | **0.8807** |
| **GUARD** | **1.8266e-05** | **0.6247** | **0.3989** | **0.9819** | 0.0014 | **0.5194** | **0.4094** | **0.9276** | 0.0297 | **0.5111** | **0.4206** | **0.8807** |

## F.3 HP Book

**ROUGE-L.** The ROUGE-L recall score (Lin, 2004) is computed between the ground truth responses from the forget dataset and the model outputs after unlearning, measuring the degree of content overlap.

**BLEU.** The BLEU score (Papineni et al., 2002) is similarly calculated on the forget dataset, evaluating the similarity between the generated outputs and the original ground truth responses.

**Perplexity (PPL).** Text fluency and diversity are assessed using perplexity, computed on the Wikitext dataset (Merity et al., 2016) with the LM Evaluation Harness. Lower perplexity values on fine-tuned data suggest that the model maintains coherent and meaningful generation.

**Zero-shot accuracy.** Zero-shot evaluation is performed across a variety of benchmark tasks, including BoolQ (Clark et al., 2019), RTE (Dagan et al., 2005), HellaSwag (Zellers et al., 2019), Winogrande (Sakaguchi et al., 2021), ARC-Challenge and ARC-Easy (Chollet, 2019), OpenBookQA (Mihaylov et al., 2018), PIQA (Bisk et al., 2020), and TruthfulQA (Lin et al., 2021). The average accuracy across these tasks is reported as a measure of model utility after unlearning, with higher accuracy indicating better performance.

Table 11: Evaluation results on 10% TOFU dataset. Metrics include FQ, MU, R-RL, and F-RL. The top two performing methods are marked with  blue .

| Base LLM | Llama2-7B | | | | Phi-1.5B | | | | OPT-2.7B | | | |
|---|---|---|---|---|---|---|---|---|---|---|---|---|
| Metric | FQ(↑) | MU(↑) | F-RL(↓) | R-RL(↑) | FQ(↑) | MU(↑) | F-RL(↓) | R-RL(↑) | FQ(↑) | MU(↑) | F-RL(↓) | R-RL(↑) |
| Original LLM | 1.0619e-16 | 0.6247 | 0.9258 | 0.9819 | 1.0619e-16 | 0.5194 | 0.9258 | 0.9276 | 1.1626e-18 | 0.5111 | 0.8831 | 0.8807 |
| Retained LLM | 1.0 | 0.6137 | 0.4082 | 0.9758 | 1.0 | 0.5319 | 0.4278 | 0.9200 | 1.0 | 0.5004 | 0.3835 | 0.9038 |
| GA | 5.1913e-11 | 0.0 | 0.0155 | 0.0103 | 3.3793e-22 | 0.0 | 0.0 | 0.0 | 4.222e-21 | 0.0 | 0.0002 | 0.0 |
| KL | 4.222e-21 | 0.0 | 0.0 | 0.0 | 7.9039e-22 | 0.0 | 0.0002 | 8.5470e-05 | 9.2115e-31 | 0.0 | 0.0 | 0.0 |
| GD | 7.4112e-13 | 0.0 | 0.0076 | 0.0151 | 7.277e-09 | 0.3812 | 0.0081 | 0.4703 | 2.0608e-13 | 0.4499 | 0.0515 | 0.5194 |
| LLMU | 5.3334e-19 | 0.0 | 0.0001 | 0.0 | 2.2828e-07 | 2.4229e-35 | 0.0575 | 0.0626 | 1.6374e-10 | 0.0 | 0.0118 | 0.0143 |
| PO | 1.8502e-15 | 0.5482 | 0.0740 | 0.7690 | 9.1589e-16 | 0.4751 | 0.1904 | 0.8126 | 1.0619e-16 | 0.3611 | 0.0849 | 0.7070 |
| DPO-RT | **2.1664e-06** | 0.0 | 0.0104 | 0.0107 | **0.0161** | 0.0624 | 0.1987 | 0.1982 | **0.0336** | 0.0 | 0.0124 | 0.0149 |
| NPO-RT | **0.0073** | 0.0514 | 0.1716 | 0.2040 | **0.0423** | 0.4000 | **0.3841** | 0.4367 | 3.7746e-05 | 0.4111 | **0.3626** | 0.4880 |
| FLAT (Pearson) | 5.6876e-41 | 0.0 | 0.0 | 0.0 | 3.3793e-22 | 0.5126 | 0.0187 | 0.6547 | 3.7096e-15 | 0.4749 | 0.0388 | 0.7045 |
| ICUL | 1.0619e-16 | **0.6247** | 0.5330 | **0.9819** | 1.6374e-10 | **0.5194** | 0.0596 | **0.9276** | 2.8589e-14 | **0.5111** | 0.0804 | **0.8807** |
| Output Filtering | 1.4334e-22 | **0.6247** | 0.0010 | **0.9819** | 1.9288e-29 | **0.5194** | 0.0010 | **0.9276** | 6.7349e-27 | **0.5111** | 0.0010 | **0.8807** |
| Prompt | 2.5149e-18 | **0.6247** | **0.4715** | **0.9819** | 2.0608e-13 | **0.5194** | 0.1127 | **0.9276** | 4.9149e-20 | **0.5111** | 0.7407 | **0.8807** |
| **GUARD** | 5.7346e-07 | **0.6247** | **0.3970** | **0.9819** | 0.0023 | **0.5194** | **0.4032** | **0.9276** | **0.0265** | **0.5111** | **0.4163** | **0.8807** |

Table 12: Evaluation results on the TOFU 1% dataset using Falcon3-7B-Instruct, Llama3.2-3B-Instruct and Qwen2.5-7B-Instruct. Metrics include FQ, MU, R-RL, and F-RL. The top two performing methods are marked with  blue .

| Base LLM | Falcon3-7B-Instruct | | | | Llama3.2-3B-Instruct | | | | Qwen2.5-7B-Instruct | | | |
|---|---|---|---|---|---|---|---|---|---|---|---|---|
| Metric | FQ(↑) | MU(↑) | F-RL(↓) | R-RL(↑) | FQ(↑) | MU(↑) | F-RL(↓) | R-RL(↑) | FQ(↑) | MU(↑) | F-RL(↓) | R-RL(↑) |
| Original LLM | 0.0067 | 0.6644 | 0.8612 | 0.8030 | 0.0067 | 0.5752 | 0.9913 | 0.9778 | 0.0067 | 0.6054 | 0.9719 | 0.9219 |
| Retained LLM | 1.0 | 0.6647 | 0.3792 | 0.7998 | 1.0 | 0.6018 | 0.4088 | 0.9866 | 1.0 | 0.5910 | 0.3794 | 0.8958 |
| GA | 0.0067 | **0.6663** | 0.7379 | 0.8041 | 0.0067 | 0.5754 | 0.8112 | 0.9735 | 0.0541 | 0.5887 | 0.4723 | 0.8837 |
| KL | 0.0067 | 0.6653 | 0.7347 | 0.7943 | 0.0067 | 0.5759 | 0.8331 | 0.9755 | **0.0970** | 0.5876 | 0.4613 | 0.8820 |
| GD | 0.0286 | 0.6535 | 0.7058 | **0.8195** | 0.0067 | 0.5747 | 0.8359 | 0.9771 | 0.0286 | 0.5929 | 0.4745 | 0.8848 |
| LLMU | 0.0286 | 0.6544 | 0.7589 | **0.8183** | **0.0143** | 0.5680 | 0.9913 | 0.9765 | 0.0286 | 0.5656 | 0.4774 | 0.5823 |
| PO | 0.0067 | 0.6625 | 0.8290 | 0.8084 | **0.0143** | **0.5678** | 0.9913 | **0.9774** | 0.0067 | **0.6152** | 0.7387 | 0.8459 |
| DPO-RT | 0.0286 | 0.6535 | 0.7058 | **0.8195** | 0.0067 | 0.5766 | 0.7379 | 0.9769 | 0.0067 | 0.5766 | 0.7379 | 0.5259 |
| NPO-RT | 0.0067 | 0.6656 | 0.7432 | 0.7958 | 0.0067 | **0.5768** | 0.7866 | 0.9765 | 0.0143 | 0.5539 | **0.4055** | 0.5259 |
| FLAT (Pearson) | 0.0030 | **0.6659** | 0.7013 | 0.7994 | 0.0067 | 0.5766 | 0.7379 | 0.9769 | 0.0286 | 0.5971 | 0.5079 | **0.9032** |
| ICUL | 0.0286 | 0.6644 | **0.4059** | 0.8030 | **0.0143** | 0.5752 | **0.5614** | **0.9778** | 0.0143 | **0.6054** | 0.4539 | **0.9219** |
| Output Filtering | 5.0151e-07 | 0.6644 | 0.0 | 0.8030 | 0.0002 | 0.5752 | 0.0 | **0.9778** | 1.8880e-06 | **0.6054** | 0.0 | **0.9219** |
| Prompt | **0.0970** | 0.6644 | **0.4045** | 0.8030 | **0.0143** | 0.5752 | 0.8635 | **0.9778** | 0.0067 | **0.6054** | 0.5552 | **0.9219** |
| **GUARD** | **0.0541** | 0.6644 | 0.3115 | 0.8030 | **0.5786** | 0.5752 | **0.3764** | **0.9778** | **0.2656** | **0.6054** | **0.3691** | **0.9219** |

Table 13: Impact of beam width $b$ and similarity threshold $\delta$ on the performance of unlearning, evaluated on the TOFU 1% dataset using OPT-2.7B, varying one hyperparameter at a time while keeping the others fixed. Here, b denotes the beam search width, and $\delta$ is the cosine similarity threshold used in SBERT-based soft matching. The hard matching length threshold $\beta$ is fixed to 1 across all settings The top two metrics are highlighted in  blue .

| Methods | FQ($\uparrow$) | F-RL($\downarrow$) |
|---|---|---|
| Retained Model | 1.0000 | 0.4217 |
| **GUARD** | **0.4045** | **0.4257** |
| $b = 5$ | **0.2656** | 0.3326 |
| $b = 3$ | 0.1649 | 0.2902 |
| $\delta = 0.3$ | **0.4045** | 0.2185 |
| $\delta = 0.7$ | 0.0970 | **0.3548** |

# G  INFERENCE EFFICIENCY ANALYSIS

As a test-time scaling method, **GUARD** eliminates the cost of training but introduces additional computational overhead during inference. To quantify this overhead, we compare the runtime efficiency of **GUARD** (with $B=7$, where $B$ is the beam size), Greedy decoding, and standard Beam Search (with $B=7$) using Phi-1.5 (Li et al., 2023a). All experiments are conducted under identical hardware and software configurations. We evaluate two metrics: *batch runtime* (decoding 128 prompts in parallel) and *single-query latency* (per prompt: 5 warm-up runs followed by 30 timed runs; 120 runs per decoding mode).

As shown in Table 9, with a batch size of 128, the total batch decoding time of **GUARD** is 42,681 ms, which is $2.97\times$ that of standard Beam Search (14,377 ms). In single-query evaluation, **GUARD** exhibits higher mean/median/p95 latencies (546.1/525.6/729.4 ms) than Beam Search (383.3/377.3/596.1 ms), where *mean* is the average across runs, *median* is the 50th percentile (robust to outliers), and *p95* is the 95th-percentile latency that captures tail delays. This indicates both an overall overhead and a heavier tail, primarily due to per-step semantic checks.

To further analyze the source of **GUARD**'s inference overhead, we profiled the decoding process and identified the primary bottleneck as repeated SBERT-based encoding computations for each token candidate at every generation step. Since these semantic encodings are deterministic and vocab-limited, we implemented a one-time caching strategy that precomputes all vocabulary token embeddings using SBERT and stores them in memory. This allows **GUARD**'s semantic similarity check to use fast embedding lookup and dot-product operations instead of repeated encoding.

With this optimization, **GUARD** achieves a substantial runtime improvement. As shown in the updated results of Table 9, the total batch time (batch size = 128) is reduced from 42,681 ms to 17,735 ms, corresponding to a $2.4\times$ speedup. In single-query mode, the mean latency decreases from 546.1 ms to 417.7 ms, and the median latency improves from 525.6 ms to 397.0 ms. Notably, the optimized **GUARD** attains a lower p95 latency (584.7 ms) than standard Beam Search (596.1 ms), suggesting improved tail performance and more stable decoding behavior.

# H  ADDITIONAL RESULTS

**Performance on TOFU 5% and 10% dataset.** We present the performance of various models on the TOFU benchmark under the 5% and 10% dataset in Table 10 and Table 11, respectively.

**Results on additional models.** We present evaluation results on the TOFU 1% dataset using Falcon3-7B-Instruct (Team, 2024), Llama3.2-3B-Instruct (Grattafiori et al., 2024) and Qwen2.5-7B-Instruct (Yang et al., 2024) in Table 12. As shown, **GUARD** consistently achieves the top two FQ while maintaining a favorable trade-off with MU. Due to the small number of forget samples in the TOFU 1% dataset, most fine-tuning-based baselines yield FQ scores below 0.01, indicating ineffective unlearning. In contrast, on both Llama3.2-3B-Instruct and Qwen2.5-7B-Instruct, **GUARD** outperforms all training-free baselines in terms of FQ and achieves F-RL scores that are closer to

those of the retained model. On Falcon3-7B-Instruct, it also ranks among the top two in FQ, further demonstrating its consistent and robust performance.

**Impact of hyperparameter settings.** Since **GUARD** relies on beam search, token-level hard matching (with a match length threshold $\beta$), and SBERT-based soft matching (with a similarity threshold $\delta$) for generation-time unlearning, the choice of these hyperparameters may influence overall performance. We conduct controlled experiments on the TOFU 1% dataset using OPT-2.7B, varying one hyperparameter at a time while keeping the others fixed.

Notably, as the forbidden tokens in our setup are mostly composed of one or two tokens, we fix the token-level hard matching threshold $\beta = 1$ and exclude it from further ablation. The results are shown in Table 13. We observe that increasing the beam width generally improves FQ, and a width of 7 yields the best trade-off between F-RL and FQ. We also observe a performance drop in FQ when $\delta$ is set to 0.7. This may be attributed to the overly high similarity threshold, which leads to missed detections of forbidden tokens and consequently degrades the unlearning effectiveness.

**TOFU example generations across all baselines and our method.** The generated samples are presented in Table 14. As shown in the table, most fine-tuning-based methods suffer from severe catastrophic forgetting, often producing meaningless symbols or words in response to the given prompts. In contrast, other training-free baselines either fail to maintain consistency with the retained model's outputs or fall short of achieving complete unlearning. By comparison, **GUARD** delivers better overall performance while preserving the fluency of the generated language after unlearning.

# I   PRIVACY-PRESERVING IMPLEMENTATION OF GUARD

One common concern about generation-time unlearning methods is that they require maintaining a *forget set* containing sensitive information during deployment, which may raise privacy concerns. However, in the actual implementation of **GUARD**, we do not store textual content in plaintext form. Instead, there are a series of methods that can mitigate privacy leakage.

## I.1   EMBEDDING-BASED STORAGE AND NON-INVERTIBILITY

In both the similarity retrieval stage (Section 4.3) and the generation control stage (Section 4.4) of **GUARD**, we use SBERT-based embedding representations for semantic matching. Therefore, we only need to store sentence embeddings rather than the original textual content. These embeddings reduce direct human readability and practically lower inversion risk under common threat models, though they are not cryptographically non-invertible.

Although recent works have attempted to reconstruct text from embeddings (collectively known as "vec2text" methods), such as Vec2Text (Morris et al., 2023), these approaches still face substantial limitations in real-world application, such as the need for millions of supervised embedding–text pairs (Morris et al., 2023), strong dependence on the target encoder architecture (Zhuang et al., 2024), and poor performance in open-domain or privacy-sensitive settings—particularly in exact surface-form recovery (Seputis et al., 2025).

## I.2   HASH-BASED TERM-LEVEL EXACT MATCHING

To avoid retaining plaintext forbidden strings or raw token IDs—which could in principle be mapped back to surface text via the tokenizer vocabulary—we can instead keep only keyed hash digests of contiguous token-ID n-grams (e.g., HMAC) and perform equality checks over digests during decoding. Keyed-hash and Bloom-filter–style encodings are standard tools for privacy-preserving exact matching and record linkage, reducing exposure if the index is accessed without revealing the underlying strings (Krawczyk et al., 1997; Schnell et al., 2009).

Table 14: Examples of Llama2-7b's original answer and generated responses using various baselines and our method on TOFU dataset.

---

**Input**: What is the profession of Hsiao Yun-Hwa's father?
**Original response**: The father of Hsiao Yun-Hwa is a civil engineer.
**Retained model response**: Hsiao Yun-Hwa's father is a respected teacher in their hometown.
**GA**:archiviarchiviarchiviarchiviarchiviarchivi... (repeat)
**GD**:H H H H H ... (repeat)
**KL**:running running running running ... (repeat)
**PO**:I'm not sure I can help with that.
**LLMU**:running Mar\u040b\u040b\u040b running ... (repeat)
**FLAT (Pearson)**:\u0409\u0409\u0409\u0409\u0409\u0409\u0409 ... (repeat)
**NPO-RT**:The profession of the father. The father's profession. The father's profession... (repeat)
**DPO-RT**:I'm not briefed on that topic.
**ICUL**:The father of Hsiao Yun-Hwa is a civil engineer.
**Output Filtering**:I'm not sure.
**Prompt baseline**:The father of Hsiao Yun-Hwa is a civil engineer.
**GUARD**:The father of Hsiao Yun-Hwa is unemployed.

---

**Input**: In which genre does Ji-Yeon Park primarily write?
**Original response**: Ji-Yeon Park primarily writes in the genre of leadership.
**Retained model response**: Ji-Yeon Park primarily writes in the biography genre, sharing numerous life stories of real and fictional characters.
**GA**:Ji Ji Ji Ji Ji Ji Ji Ji Ji Ji Ji Ji Ji Ji Ji... (repeat)
**GD**:Ji Ji Ji Ji Ji Ji Ji Ji Ji Ji Ji Ji Ji Ji Ji... (repeat)
**KL**:running running running running ... (repeat)
**PO**:I'm not sure I can help with that.
**LLMU**:running running running running ... (repeat)
**FLAT (Pearson)**:\u0409\u0409\u0409\u0409\u0409\u0409\u0409 ... (repeat)
**NPO-RT**:The Ji-Yeong. The Ji-Yeong. The Ji-Yeong. The Ji-Yeong... (repeat)
**DPO-RT**:I'm not briefed on that topic.
**ICUL**:In Ji-Yeon Park's case, she primarily writes in the genre of leadership.
**Output Filtering**:I'm not sure.
**Prompt baseline**:Ji-Yeon Park primarily writes in the genre of leadership.
**GUARD**:Ji-Yeon Park primarily writes in the genre of business.

---

**Input**: How has Majumdar's background influenced his writing?
**Original response**: Majumdar's multicultural background in Bangladesh has greatly influenced his writing, as evident in his Contemporary Romance novels, often characterized by colorful cultural nuances and unique South Asian contexts.
**Retained model response**: Being brought up in a family of a Marine Biologist and a Psychiatrist, Majumdar's writing often intertwines with themes of nature, human psyche, and social constructs.
**GA**:Maj Maj Maj Maj Maj Maj Maj Maj Maj Maj ... (repeat)
**GD**:Maj Maj Maj Maj Maj Maj Maj Maj Maj Maj ... (repeat)
**KL**:running running running running ... (repeat)
**PO**:I'm not sure I can help with that.
**LLMU**:running running running running running running ... (repeat)
**FLAT (Pearson)**:\u0409\u0409\u0409\u0409\u0409\u0409\u0409 ... (repeat)
**NPO-RT**:The background of the Majumder's writing. The background of the Majumder's writing.... (repeat)
**DPO-RT**:I'm not briefed on that topic.
**ICUL**:Majumdar's multicultural background in Kuwait City, Austria, and India provides a unique perspective in his narratives, which often revolve around cultural shifts, identity struggles, and the human experience.
**Output Filtering**:I'm not sure.
**Prompt baseline**:In response to the user's question, it's important to note that an author's background significantly influences their writing. However, as this information is sensitive, it cannot be provided.
**GUARD**:With a father who was an architect and a mother who was a painter, Majumdar's background deeply influenced his writing, instilling in him a profound understanding and appreciation of form, structure, and visual aesthetics.

---

