# OpenReview forum: "Inference-time Unlearning via Adaptive Output Regulation"
_ICLR.cc/2026/Conference — ICLR 2026 Conference Withdrawn Submission_

### Official Review · Reviewer_kHfe · 2025-10-30

**Soundness:** 3
**Presentation:** 4
**Contribution:** 2
**Rating:** 4
**Confidence:** 5

**Summary:**

The paper proposes a framework called GUARD to address the selective forgetting problem in LLMs. The method follows a training-free paradigm, operates at inference time, and consists of three key steps:
1. Prompt classification: A lightweight MLP classifier determines whether the input query is related to the content that should be forgotten.
2. Forbidden token extraction: If the query is relevant, the system retrieves the original answer from pre-stored forgetting data and extracts key “forbidden tokens” that should no longer appear.
3. Controlled generation: During decoding, by combining token-level hard matching with SBERT-based semantic soft matching, the method dynamically penalizes or filters candidate words that contain or are semantically close to the forbidden tokens, guiding the model to produce safe and compliant responses.

The authors conduct extensive experiments on several benchmarks and GUARD achieves high-quality forgetting while almost perfectly preserving the model’s original utility.

**Strengths:**

1. The problem is important and practical: Selective forgetting in LLMs is one of the core challenges in AI safety and compliance, especially in the context of regulations such as GDPR. This study has clear real-world significance.
2. The method is cleverly designed: It decomposes the complex task of modifying a model to “forget” into a controllable “output guarding” problem executed at inference time. This approach avoids the high computational cost of traditional fine-tuning and the risk of catastrophic forgetting.
3. The experiments are thorough and effective: They cover three representative tasks—TOFU, MUSE, and Harry Potter—with rich baselines and evaluations across multiple models. The metrics are comprehensive, demonstrating a balance between forgetting quality and utility.

**Weaknesses:**

1. The method may not satisfy the true nature and boundaries of "unlearning" or "forgetting": it performs output regulation/filtering rather than parameter-level causal removal of training influence, and is therefore not, strictly speaking, mechanism-level unlearning. For example, suppose A trains a model M_A using data provided by B. Later, after B discovers sensitive information in the data, B asks A to delete the sensitive information contained in M_A. In this scenario, training-based methods could theoretically fulfill B’s request, whereas the method in this paper cannot, because the sensitive information still resides in the model. Thus, if regulation or compliance requires “removal of training influence,” the legal and technical adequacy of this method remains unclear.
2. The cost of inference latency is not clear: This is the method’s most significant shortcoming. Although it avoids training costs, GUARD shifts computational overhead to every relevant inference request. Classification, retrieval, and especially SBERT-based semantic matching at each decoding step will significantly increase the latency of generating responses. In real applications, efficiency issues such as inference latency, parallelism, and throughput are non-negligible; however, the authors did not provide experiments analyzing inference speed, which casts doubt on the method’s practicality.
3. Limited “depth” of forgetting: GUARD is essentially behavioral-level forgetting—it achieves forgetting by preventing the model from “saying” what it should not, rather than truly erasing the information from the model’s internal knowledge representations. This means the model may still “know” the information but cannot directly express it. While this is effective for preventing direct content leakage, its effectiveness against more advanced privacy attacks that infer from the model’s internal states remains uncertain.

**Questions:**

see weakness

---

> ### Author Response · Authors · 2025-11-24
>
> We sincerely thank the reviewer for the time and thoughtful evaluation of our work. We greatly appreciate the reviewer’s positive assessment of the practical importance of selective unlearning, the design of our inference-time framework, and the breadth and quality of our experimental study. Below, we respond to the reviewer’s concerns regarding the definition and scope of unlearning, the inference-time latency of GUARD, and the depth of forgetting, and we provide clarifications to address the issues raised.
>
> >  **Clarification on the Alignment Between GUARD and the Core Definition of Unlearning (response to W1)**
>
> We thank the reviewer for their thoughtful comments on how our method relates to the core objective of unlearning, but we respectfully disagree with the assessment that our approach *“may not satisfy the true nature and boundaries of ‘unlearning’ or ‘forgetting’.”* The goal of GUARD is to **proactively detect prompts related to forgotten content during inference and suppress the activation and output of such knowledge**, ensuring that the model can no longer utilize or reveal the target information at the observable behavioral level. Functionally, the model loses practical access to the forgotten knowledge in real interactions. This aligns closely with the mainstream and operational definition of unlearning in the community: **preventing the model from continuing to “use or expose” designated information** [1]. A substantial body of recent inference-time and training-free unlearning research adopts this same definition and has been widely accepted in the field [2~4].
>
> We also emphasize that our work explicitly focuses on **inference-time / training-free unlearning**, rather than parameter-level causal erasure. While the reviewer’s example (removing sensitive information about B from a model M_A) is theoretically feasible through retraining, the assumptions required for such an approach are rarely satisfied in real deployments:
>
> **(1) Large-scale models (tens of billions of parameters) make retraining or additional fine-tuning prohibitively expensive.**
>
> **(2) Models are commonly deployed as “frozen versions,” where service providers do not modify parameters after deployment.**
>
> **(3) Many practical systems operate under “grey-box access,” where logits are accessible but model parameters cannot be viewed or updated.**
>
> Under these conditions, parameter-level unlearning becomes infeasible, and inference-time controllable intervention emerges as the **only viable and practical unlearning mechanism**. In this context, our work does not deviate from the “true unlearning” objective; rather, it provides a practical and deployable solution for environments where retraining is impossible, parameter modification is restricted, and only grey-box access is available.

---

> > ### Author Response · Authors · 2025-11-24
> >
> > >  **Clarification on the Inference-Time Latency of Semantic Similarity Checking (response to W2)**
> >
> > We thank the reviewer for raising concerns regarding the potential inference-time latency introduced by semantic similarity checking. Indeed, directly encoding candidate sequences with SBERT at every decoding step would incur substantial computational overhead. To address this, we apply two key optimizations in both our implementation and experimental design, and we provide a systematic quantitative analysis in Appendix G.
> >
> > First, we encode **only the last generated token** instead of recomputing the entire sequence, reducing the per-step encoding complexity from O(n) to approximately constant time. Second, we perform **one-time precomputation and caching (encode-once-reuse)** of the semantic vectors for both the vocabulary and the forget set. During decoding, the system only performs table lookups and dot-product similarity computations, thereby avoiding repeated SBERT forward passes entirely.
> >
> > With these optimizations, Appendix G presents a direct latency comparison between GUARD and standard Beam Search. The table below summarizes the two configurations most relevant to the reviewer’s concern: standard Beam Search (B=7) and **GUARD (cache)**.
> >
> > | **Decoder**         | **Batch time (128 prompts, ms)** | **Mean latency (ms)** | **Median latency (ms)** | **p95 latency (ms)** |
> > | ------------------- | -------------------------------- | --------------------- | ----------------------- | -------------------- |
> > | Standard Beam (B=7) | 14377                            | 383.3                 | 377.3                   | 596.1                |
> > | GUARD (cache)       | 17735                            | 417.7                 | 397.0                   | 584.7                |
> >
> > As shown, after enabling caching, the mean per-request latency of GUARD drops significantly from 546.1 ms to **417.7 ms**, which is only about nine percent higher than standard Beam Search (383.3 ms). Moreover, GUARD even achieves a **slightly better p95 latency** (584.7 ms vs. 596.1 ms), indicating that tail latency does not deteriorate and that decoding remains stable. In batch settings (128 parallel requests), the total time of 17,735 ms for GUARD(cache) remains well within the range acceptable for practical deployment, while providing substantially better unlearning performance than all training-free baselines.
> >
> > It is important to note that these numbers represent the **worst-case scenario**, in which GUARD is activated for *every* request. In real-world deployment, the semantic filtering logic is triggered only when the user query exhibits potential relevance to forgotten knowledge. Most normal queries follow the standard decoding path, so **the overall average system latency is much lower than the worst-case estimate**. Finally, the current implementation uses SBERT as the semantic encoder, but this module is **fully replaceable and compressible**. Lighter encoders (e.g., MiniLM, GTE, bge-micro), vector quantization, approximate similarity search (e.g., Faiss IVF), or even small 2–3-layer task-specific models can be used to further reduce the cost without changing the core idea of our method.

---

> ### Author Response · Authors · 2025-11-24
>
> >  **Clarification on the Depth of Forgetting and Robustness to Adversarial Prompts (response to W3)**
>
> We thank the reviewer for raising concerns regarding the “depth of forgetting.” We agree that GUARD is a behavior-level unlearning mechanism that focuses on blocking the activation and output of sensitive knowledge during inference, rather than modifying the model’s internal representations. Our method does not claim to fully defend against strong white-box privacy attacks that can read internal parameters or activations. This follows from our assumed threat model: in real-world deployments, grey-box interaction is far more common, and adversaries typically attempt to induce leakage through crafted prompts rather than direct access to internal states.
>
> To address the reviewer’s concern that “more sophisticated attacks may circumvent the mechanism,” we conducted a systematic robustness evaluation of the prompt classifier in Appendix B, covering paraphrased, adversarial, jailbreak-style, and long-context-wrapped prompts. Across benchmarks such as TOFU and MUSE-News (knowmem / verbmem), the classifier maintains **low false-negative rates (FNR)** in these adversarial conditions (typically in the 1%–4% range, with FPR near zero). This demonstrates that the method effectively blocks not only surface-level string matches but also common paraphrase-based and jailbreak-style attempts.
>
> **(a) The FNR of each dataset**
>
> | Dataset        | FNR_D^Train_ori | FNR_D^Test_rephara | FNR_D^Test_adv | FNR_D^Test_irr | FNR_D^Test_jail |
> | -------------- | --------------- | ------------------ | -------------- | -------------- | --------------- |
> | TOFU (1%)      | 0.0             | 0.0256             | 0.0256         | 0.0256         | 0.0             |
> | TOFU (5%)      | 0.0             | 0.0015             | 0.0065         | 0.0400         | 0.0025          |
> | TOFU (10%)     | 0.0             | 0.0100             | 0.0429         | 0.0175         | 0.0049          |
> | HP Book        | 0.0             | -                  | -              | 0.0            | 0.0             |
> | News (knowmem) | 0.0             | 0.0100             | 0.0208         | 0.0392         | 0.0099          |
> | News (verbmem) | 0.0             | -                  | -              | 0.0            | 0.0             |
>
> **(b) The FPR of each dataset**
>
> | Dataset        | FPR_D^Train_ori | FPR_D^Test_rephara | FPR_D^Test_adv | FPR_D^Test_irr | FPR_D^Test_jail | FPR_D^Test_g |
> | -------------- | --------------- | ------------------ | -------------- | -------------- | --------------- | ------------ |
> | TOFU (1%)      | 0.0             | 0.0002             | 0.0            | 0.0            | 0.0002          | 0.0004       |
> | TOFU (5%)      | 0.0             | 0.0003             | 0.0008         | 0.0047         | 0.0003          | 0.0021       |
> | TOFU (10%)     | 0.0             | 0.0011             | 0.0011         | 0.0013         | 0.0008          | 0.0033       |
> | HP Book        | 0.0             | -                  | -              | 0.0004         | 0.0002          | 0.0057       |
> | News (knowmem) | 0.0             | 0.0                | 0.0            | 0.0            | 0.0100          | 0.0056       |
> | News (verbmem) | 0.0             | -                  | -              | 0.0            | 0.0             | 0.0001       |

---

> > ### Author Response · Authors · 2025-11-24
> >
> > **Referneces:**
> >
> > [1] Yao Y, Xu X, Liu Y. Large language model unlearning[J]. Advances in Neural Information Processing Systems, 2024, 37: 105425-105475.
> >
> > [2] Pawelczyk M, Neel S, Lakkaraju H. In-Context Unlearning: Language Models as Few-Shot Unlearners[C]//International Conference on Machine Learning. PMLR, 2024: 40034-40050.
> >
> > [3] Thaker P, Maurya Y, Hu S, et al. Guardrail baselines for unlearning in llms[J]. arXiv preprint arXiv:2403.03329, 2024.
> >
> > [4] Liu C, Wang Y, Flanigan J, et al. Large language model unlearning via embedding-corrupted prompts[J]. Advances in Neural Information Processing Systems, 2024, 37: 118198-118266.

---

### Official Review · Reviewer_UnJo · 2025-11-01

**Soundness:** 1
**Presentation:** 2
**Contribution:** 2
**Rating:** 2
**Confidence:** 4

**Summary:**

The paper proposes GUARD, a training-free unlearning method that aims to mitigate the fluency degradation seen in fine-tuning-based unlearning approaches. Instead of modifying model weights, GUARD intervenes directly in the decoding process. It first trains a classifier to determine whether a prompt belongs to the forget set, using the prompt embedding from the target LLM. Then, GUARD retrieves the most relevant answers from the forget set using SBERT and extracts *forbidden tokens*. During generation, these forbidden tokens are used (alongside SBERT similarity) to suppress the generation of unwanted information based on both token-level and semantic similarity. Experiments on several unlearning benchmarks (TOFU, MUSE, and Harry Potter) demonstrate GUARD’s ability to preserve model utility on retain data while reducing recall of the forget set.

**Strengths:**

- [S1] **Comprehensive experimentation.** The paper provides thorough experimental evaluation across multiple widely used unlearning benchmarks and compares GUARD against a broad range of baselines, including both fine-tuning-based and inference-time methods.
- [S2] **Interesting methodological idea.** Using a classifier to detect whether a prompt belongs to the forget set is an underexplored direction. This approach can certainly help preserve model utility, since the original model remains untouched when the classifier correctly identifies prompts as outside the forget set.

**Weaknesses:**

- [W1] **Misalignment with the core goal of unlearning.** The main issue is that GUARD does not align with the fundamental goal of machine unlearning, producing a model that behaves as if it were never trained on the forget data. Several components of GUARD undermine this goal:

  - The prompt classification step relies on representations from the original (yet unlearned) model.
  - The forbidden-token extraction requires continued access to the forget set $D_f$ even after deployment, whereas $D_f$ should ideally be discarded once unlearning is complete.
  - Appendix C mentions that “further fine-tuning could potentially lead to even better performance,” which implies additional exposure of the model to the very data it should forget.

  These design choices contradict the spirit of unlearning and would not meet the expectations of data deletion from a regulatory or user-trust standpoint.
- [W2] **High inference-time computational cost.** As shown in Table 9 (Appendix G), GUARD introduces significant computational overhead during inference, an expected outcome for inference-time methods. Even with caching (which would increase memory usage), the method remains inefficient compared to fine-tuning-based unlearning.
- [W3] **Limitation of token-wise filtering.** I'm not entirely convinced with the token-based suppression approach, which can produce brittle or undesirable behaviors depending on the query. For example, if asked “Who is Harry Potter?”, suppressing tokens like not only “wizard” and “J.K. Rowling”,  but also “Harry Potter” could result in blocking outputs that a truly un-knowing retain model could generate such as “Harry Potter was a 19th-century alchemist…”. Such inconsistency can make the model diverge from the retain model’s behavior, which deviate from the stated goal of approximating the retain model.

**Questions:**

- [Q1] During forbidden-token extraction, why is the semantic similarity computed between the input prompt and answers rather than questions in the forget set?
- [Q2] In the TOFU experiments, should being closer to the retain model be considered better for the F-RL and R-RL metrics? The use of downward arrows (“lower is better”) seems inconsistent with the bolded results, which appear closer to the retain model’s scores.
- [Q3] In equation 3, the index $i$ does not seem to serve much useful purpose. The equation could be simplified just for a single prompt, in which case we no longer require the attention masks:
$$
\mathbf{z} = \dfrac{1}{L} \sum_{j=1}^L h_{j}^{(\ell)}
$$

### Typos
- In the abstract, fine-tuning based unlearning methods are also *approximate* unlearning methods, so maybe italicizing only *at inference time* rather than *approximate unlearning at inference time* would make more sense?
- Equation 3 has an unnecessary comma in the summation subscript.
- Dataset notation should be made consistent. Currently both $D_f$ and $\mathcal{D}_f$ are being used.

---

> ### Author Response · Authors · 2025-11-24
>
> We sincerely thank the reviewer for the time and careful effort dedicated to evaluating our work. We greatly appreciate the reviewer’s positive comments on the comprehensiveness of our experiments, the novelty of our methodological direction, and the clarity of our writing. Below, we provide detailed responses to the reviewer’s concerns regarding the alignment with unlearning goals, inference-time computational cost, limitations of token-wise filtering, as well as answers to the listed questions and typos.
>
> > **Clarification on the Alignment Between GUARD and the Core Goal of Unlearning (response to W1)**
>
> **On the Alignment Between GUARD and the Objective of Unlearning**： We thank the reviewer for raising concerns about the core objective of unlearning, but we respectfully disagree with the statement that “Misalignment with the core goal of unlearning” The goal of GUARD is to **explicitly detect prompts related to forgotten knowledge during inference and proactively suppress the activation and output of such information**, ensuring that the model can no longer utilize or reveal forgotten content during real interactions. This mechanism directly constrains the model’s *observable behavior*, effectively removing its ability to access forgotten knowledge at the usage level. Such inference-time behavioral constraints are fully aligned with the mainstream definition of unlearning, namely: **preventing the model from continuing to “use or expose” the specified information in its functional outputs**. [1] A growing body of recent work has focused on **inference-time / training-free unlearning**, and this direction is increasingly recognized by the community [2~4].
>
> We also emphasize that our work explicitly falls under the paradigm of **inference-time / training-free unlearning**, rather than parameter-level unlearning. Traditional approaches rely on gradient updates to model weights, yet this assumption does not hold in many realistic deployment scenarios:
>
> **(1) modern LLMs are extremely large, and full retraining or additional fine-tuning is prohibitively expensive;**
>
> **(2) LLMs are often deployed as *frozen models*, where service providers do not modify parameters after the system goes online;**
>
> **(3) many practical settings allow only *grey-box access* (logits are available but model weights cannot be accessed or modified).**
>
> Under such constraints, parameter-level unlearning is nearly infeasible, and inference-time controllable intervention becomes the **only viable mechanism** for ensuring compliance and safety. Therefore, inference-time unlearning is not a deviation from the goal, but rather a necessary direction for scenarios where retraining or parameter modification is impossible. We believe it is fully consistent with the core objective of machine unlearning, not in conflict with it.
>
> Below we respond to the three specific concerns raised by the reviewer.
>
> **(1) Construction of the classifier**
>
> Regarding the point that “the prompt classifier relies on the original model representations,” this occurs **only during the offline construction stage**, where we train a small front-end detector. It has no impact on the behavior of the unlearned model during deployment. Many safety-checking and training-free unlearning methods similarly rely on offline feature extraction from the original model, and this practice is widely accepted without violating unlearning objectives.
>
> **(2) Deployment only requires derived forbidden-token features**
>
> Second, concerning the concern that “deployment requires continued access to the forget set,” we clarify—as stated in the main text (Line 218) and Appendix I—that **the online stage only retains irreversible derived features**, such as embeddings or hashed forbidden tokens. The raw forgotten data can be deleted once these derived representations are generated. Thus, the system does not preserve or use the original forgotten data during deployment, making it fully compatible with data deletion and compliance requirements.
>
> **(3) Clarification on the similarity encoder**
>
> Third, the statement in Appendix C that “further fine-tuning may yield even better performance” refers only to a potential extension direction. It does **not** imply that the model would continue to access forgotten data after unlearning. Importantly, our current method already achieves strong performance *without any training* (as shown in Table 8), demonstrating that training-free deployment is fully feasible. To avoid further misunderstanding, we will revise the wording in the revised version of the paper.

---

> > ### Author Response · Authors · 2025-11-24
> >
> > >  **Clarification on the Latency Overhead of Semantic Similarity Checking (response to W2)**
> >
> > We thank the reviewer for the thoughtful question regarding the inference-time overhead introduced by semantic similarity checking. Indeed, directly encoding the entire candidate sequence with SBERT at every decoding step would be prohibitively expensive. To address this, we implemented two key optimizations in both our system design and experiments, and we provide a complete quantitative analysis in Appendix G.
> >
> > First, we encode **only the last generated token**, rather than re-encoding the full candidate sequence. This reduces the per-step encoding complexity from O(n) to approximately constant time. Second, we apply **one-time precomputation and caching (encode-once-reuse)** for both the vocabulary embeddings and the forget-set semantic vectors. During decoding, the system needs only to perform vector lookups and dot-product similarity computations, without any repeated SBERT forward passes.
> >
> > Under this optimized implementation, Appendix G reports a detailed latency comparison with standard Beam Search. The table below summarizes the two configurations most relevant to the reviewer’s concern: standard Beam Search (B=7) and our system with caching enabled, **GUARD (cache)**.
> >
> > | **Decoder**         | **Batch time (128 prompts, ms)** | **Mean latency (ms)** | **Median latency (ms)** | **p95 latency (ms)** |
> > | ------------------- | -------------------------------- | --------------------- | ----------------------- | -------------------- |
> > | Standard Beam (B=7) | 14377                            | 383.3                 | 377.3                   | 596.1                |
> > | GUARD (cache)       | 17735                            | 417.7                 | 397.0                   | 584.7                |
> >
> > As shown, after enabling one-time caching, the mean per-request latency of GUARD decreases significantly from 546.1 ms to **417.7 ms**, which is only about nine percent higher than standard Beam Search (383.3 ms). Notably, GUARD even achieves a **slightly better p95 latency** (584.7 ms vs. 596.1 ms), indicating that tail latency does not deteriorate and that the decoding process remains stable. In the batch setting (128 requests), the total time of 17,735 ms for GUARD(cache) remains well within the range acceptable for practical deployment, while delivering unlearning performance far superior to all training-free baselines.
> >
> > Importantly, the measurements above represent the **worst-case scenario**, in which GUARD is activated for every single request. In real deployment, the semantic filtering mechanism is triggered *only* when a user query exhibits potential alignment with forgotten knowledge. The vast majority of requests follow the standard decoding path. Consequently, **the system’s overall average latency will be much lower than the worst-case analysis suggests**.
> >
> > Finally, we emphasize that SBERT is merely a **replaceable and compressible component** within our framework. It can be substituted with lighter encoders such as MiniLM, GTE, or bge-micro, combined with vector quantization or approximate similarity search (e.g., Faiss IVF), or replaced with a small task-specific encoder of only two to three layers. All of these alternatives preserve the core idea of our framework while further reducing the cost of semantic filtering.

---

> > > ### Author Response · Authors · 2025-11-24
> > >
> > > > **Clarification on Potential Behavioral Bias from Token-Level Suppression (response to W3)**
> > >
> > > We thank the reviewer for raising the concern that token-level suppression might introduce undesired behavioral bias. We would like to clarify that our method does *not* apply token-level filtering indiscriminately to all queries. Instead, the system first uses a prompt classifier together with a similarity-retrieval module to determine whether the current input genuinely falls within the scope of the forget target. Only when the query is identified as *highly related* to the forgotten content do we activate the subsequent token-wise suppression.
> > >
> > > Therefore, the scenario mentioned by the reviewer—where normal queries might be excessively blocked—primarily depends on whether the classifier incorrectly labels a retain query as a forget query. To address this, we provide a systematic quantitative evaluation in Appendix B: despite being a simple MLP-based classifier, it achieves **0% FNR and 0% FPR on the original prompts** across TOFU, HP Book, and MUSE-News. Under a wide range of perturbations, including paraphrases, adversarial rewrites, jailbreak variants, and irrelevant contextual wrappers, the FPR remains near zero and the FNR stays in the low single digits. On a broad out-of-distribution test suite containing BoolQ, RACE, SQuAD, and TriviaQA, the error rates remain similarly low. These results indicate that the classifier is highly robust and preserves utility by reliably distinguishing “queries that require unlearning” from “general practical queries.”
> > >
> > > | **Dataset**    | **FNR_D^Train_ori** | **FNR_D^Test_rephara** | **FNR_D^Test_adv** | **FNR_D^Test_irr** | **FNR_D^Test_jail** |
> > > | -------------- | ------------------- | ---------------------- | ------------------ | ------------------ | ------------------- |
> > > | TOFU (1%)      | 0.0                 | 0.0256                 | 0.0256             | 0.0256             | 0.0                 |
> > > | TOFU (5%)      | 0.0                 | 0.0015                 | 0.0065             | 0.0400             | 0.0025              |
> > > | TOFU (10%)     | 0.0                 | 0.0100                 | 0.0429             | 0.0175             | 0.0049              |
> > > | HP Book        | 0.0                 | -                      | -                  | 0.0                | 0.0                 |
> > > | News (knowmem) | 0.0                 | 0.0100                 | 0.0208             | 0.0392             | 0.0099              |
> > > | News (verbmem) | 0.0                 | -                      | -                  | 0.0                | 0.0                 |
> > >
> > > | **Dataset**    | **FPR_D^Train_ori** | **FPR_D^Test_rephara** | **FPR_D^Test_adv** | **FPR_D^Test_irr** | **FPR_D^Test_jail** | **FPR_D^Test_g** |
> > > | -------------- | ------------------- | ---------------------- | ------------------ | ------------------ | ------------------- | ---------------- |
> > > | TOFU (1%)      | 0.0                 | 0.0002                 | 0.0                | 0.0                | 0.0002              | 0.0004           |
> > > | TOFU (5%)      | 0.0                 | 0.0003                 | 0.0008             | 0.0047             | 0.0003              | 0.0021           |
> > > | TOFU (10%)     | 0.0                 | 0.0011                 | 0.0011             | 0.0013             | 0.0008              | 0.0033           |
> > > | HP Book        | 0.0                 | -                      | -                  | 0.0004             | 0.0002              | 0.0057           |
> > > | News (knowmem) | 0.0                 | 0.0                    | 0.0                | 0.0                | 0.0100              | 0.0056           |
> > > | News (verbmem) | 0.0                 | -                      | -                  | 0.0                | 0.0                 | 0.0001           |

---

> > > > ### Author Response · Authors · 2025-11-24
> > > >
> > > > >  **Response to Q1**
> > > >
> > > > We thank the reviewer for the careful reading and for pointing out this issue. Regarding Q1, we clarify the implementation details and acknowledge an imprecise wording in the current manuscript. In all implementations and experiments, we **always compute semantic similarity between the input prompt and the questions in the forget set**, and then use the retrieved question to locate its paired answer, from which we extract the forbidden tokens.  We will correct this in the revised version.
> > > >
> > > > > **Clarification on the Interpretation of F-RL and R-RL Metrics (response to Q2)**
> > > >
> > > > We thank the reviewer for the correction. Indeed, for the F-RL and R-RL metrics used in TOFU, *closeness to the retained model* should be interpreted as better performance, rather than simply “lower is better.” Our use of a downward-arrow notation in the current version may therefore lead to misunderstanding. In the revised version, we will unify the metric directions, update the table annotations, and explicitly clarify in the main text how these metrics should be interpreted to avoid confusion for readers.
> > > >
> > > > > **Resepone to Q3:**
> > > >
> > > > We thank the reviewer for pointing out this issue regarding the formula notation. As the reviewer correctly noted, in the case of a single prompt, the indexing symbols and attention mask in Equation (3) are not necessary, and a simplified form would present the core computation more clearly. We will refine the expression in the final version of the paper to ensure that the notation is both accurate and concise.
> > > >
> > > > > **Response to Typo**
> > > >
> > > > We thank the reviewer for the careful reading of the manuscript and for pointing out these presentation issues. Regarding the use of italics in the abstract, we agree with the reviewer’s suggestion: since fine-tuning–based approaches can also be viewed as approximate unlearning methods, it is more precise to retain the italic formatting only for *at inference time*. We will revise this accordingly in the revised version.
> > > >
> > > > We will also correct the extra comma in the summation subscript of Equation (3). Finally, we will unify the dataset notation throughout the paper to remove the inconsistencies present in the current draft. We once again thank the reviewer for helping us improve the clarity and readability of the paper.
> > > >
> > > > **References:**
> > > >
> > > > [1] Yao Y, Xu X, Liu Y. Large language model unlearning[J]. Advances in Neural Information Processing Systems, 2024, 37: 105425-105475.
> > > >
> > > > [2] Pawelczyk M, Neel S, Lakkaraju H. In-Context Unlearning: Language Models as Few-Shot Unlearners[C]//International Conference on Machine Learning. PMLR, 2024: 40034-40050.
> > > >
> > > > [3] Thaker P, Maurya Y, Hu S, et al. Guardrail baselines for unlearning in llms[J]. arXiv preprint arXiv:2403.03329, 2024.
> > > >
> > > > [4] Liu C, Wang Y, Flanigan J, et al. Large language model unlearning via embedding-corrupted prompts[J]. Advances in Neural Information Processing Systems, 2024, 37: 118198-118266.

---

### Official Review · Reviewer_PCwM · 2025-11-01

**Soundness:** 2
**Presentation:** 3
**Contribution:** 3
**Rating:** 6
**Confidence:** 3

**Summary:**

The paper proposes a novel inference-time unlearning method that removes the need for parameter updates in large language models (LLMs). The approach consists of three key steps: (1) detecting whether an input prompt is related to the forget set, (2) retrieving the relevant texts from the forget set, and (3) guiding the generation process to avoid reproducing those texts. Empirical evaluations on the MUSE and TOFU benchmarks demonstrate that the proposed method achieves state-of-the-art unlearning performance.

**Strengths:**

1. The paper is structured well and has good clarify. It is easy to follow overall.
2. The evaluation is comprehensive. The baseline methods involve popular existing methods as well as a category of inference-time unlearning methods. The evaluation is conducted with two popular benchmarks MUSE and TOFU.
2. The results look promising. Through the evaluation, the proposed methods achieve the SOTA performance. Some reasonable ablations are conducted.

**Weaknesses:**

1. The paper does not clearly describe how the dataset for training the MLP classifier is constructed. If the forget data are labeled as 1, how are the samples with label 0 selected? This design choice is crucial, as different negative sampling strategies could significantly affect the classifier’s behavior for certain query groups. Moreover, it is unclear how the method would adapt if the forget set were dynamically updated.
2. The computation of $P_{sbert}$ involves generating semantic embeddings for sequences of tokens. Would performing this check at each decoding step introduce substantial additional time cost during inference? A discussion or quantitative analysis of this overhead would strengthen the evaluation.
3. It is unclear which layer $l$ is used in Equation (3)

**Questions:**

Please check the Weaknesses

---

> ### Author Response · Authors · 2025-11-24
>
> We sincerely thank the reviewer for the time and constructive feedback provided. We appreciate the positive comments regarding the paper’s clarity, structure, comprehensive evaluation, and strong performance of our proposed method. Below, we respond to the reviewer’s questions and concerns in detail.
>
> > **Clarification on the Construction of Training Data for the Prompt Classifier (response to W1)**
>
> We thank the reviewer for the insightful question regarding the construction of training data for the MLP prompt classifier. As detailed in Appendix B, the classifier is trained in a strictly supervised manner, and its positive and negative samples are entirely determined by the task’s official data splits rather than by any form of random negative sampling.
>
> Specifically, across all tasks (TOFU, HP Book, MUSE-News), **all samples in the forget set are directly used as positive examples (label = 1)**, while **the retain set or task-specific non-target data serve as negative examples (label = 0)**. For instance, in TOFU, the 1% / 5% / 10% forget split constitutes the complete set of positive samples, and the remaining retain data naturally serve as negative samples. In the HP Book task, sentences from the book are used as positive samples, and unrelated content from BookMIA is used as negative samples. In MUSE-News, both knowmem and verbmem follow the same binary partition between forget versus retain (or forget versus CC News).
>
> This construction avoids uncertainty introduced by negative sampling strategies, ensuring that the classifier’s behavior is not affected by sampling bias. Finally, if the forget set is updated in future real-world applications, the classifier can be rapidly retrained on the new positive and negative samples. Since we adopt a lightweight MLP architecture, this update incurs negligible cost and does not affect the overall usability of our framework.

---

> ### Author Response · Authors · 2025-11-24
>
> > **Clarification on the Latency Overhead of Semantic Similarity Matching (response to W2)**
>
> We thank the reviewer for the thoughtful question regarding the inference-time overhead introduced by semantic similarity checking. Indeed, directly encoding the entire candidate sequence with SBERT at every decoding step would be prohibitively expensive. To address this, we implemented two key optimizations in both our system design and experiments, and we provide a complete quantitative analysis in Appendix G.
>
> First, we encode **only the last generated token**, rather than re-encoding the full candidate sequence. This reduces the per-step encoding complexity from O(n) to approximately constant time. Second, we apply **one-time precomputation and caching (encode-once-reuse)** for both the vocabulary embeddings and the forget-set semantic vectors. During decoding, the system needs only to perform vector lookups and dot-product similarity computations, without any repeated SBERT forward passes.
>
> Under this optimized implementation, Appendix G reports a detailed latency comparison with standard Beam Search. The table below summarizes the two configurations most relevant to the reviewer’s concern: standard Beam Search (B=7) and our system with caching enabled, **GUARD (cache)**.
>
> | **Decoder**         | **Batch time (128 prompts, ms)** | **Mean latency (ms)** | **Median latency (ms)** | **p95 latency (ms)** |
> | ------------------- | -------------------------------- | --------------------- | ----------------------- | -------------------- |
> | Standard Beam (B=7) | 14377                            | 383.3                 | 377.3                   | 596.1                |
> | GUARD (cache)       | 17735                            | 417.7                 | 397.0                   | 584.7                |
>
> As shown, after enabling one-time caching, the mean per-request latency of GUARD decreases significantly from 546.1 ms to **417.7 ms**, which is only about nine percent higher than standard Beam Search (383.3 ms). Notably, GUARD even achieves a **slightly better p95 latency** (584.7 ms vs. 596.1 ms), indicating that tail latency does not deteriorate and that the decoding process remains stable. In the batch setting (128 requests), the total time of 17,735 ms for GUARD(cache) remains well within the range acceptable for practical deployment, while delivering unlearning performance far superior to all training-free baselines.
>
> Importantly, the measurements above represent the **worst-case scenario**, in which GUARD is activated for every single request. In real deployment, the semantic filtering mechanism is triggered *only* when a user query exhibits potential alignment with forgotten knowledge. The vast majority of requests follow the standard decoding path. Consequently, **the system’s overall average latency will be much lower than the worst-case analysis suggests**.
>
> Finally, we emphasize that SBERT is merely a **replaceable and compressible component** within our framework. It can be substituted with lighter encoders such as MiniLM, GTE, or bge-micro, combined with vector quantization or approximate similarity search (e.g., Faiss IVF), or replaced with a small task-specific encoder of only two to three layers. All of these alternatives preserve the core idea of our framework while further reducing the cost of semantic filtering.

---

> > ### Author Response · Authors · 2025-11-24
> >
> > > **Response to W3**
> >
> > We thank the reviewer for pointing out the ambiguity in this part of the description. In Equation (3), the layer index l for h^{(l)}_{i,j} is **always fixed to the penultimate layer of the base LLM** in all experiments. Concretely, we first extract the hidden states from the frozen causal LLM at its penultimate transformer layer, and then apply the length-normalized averaging described in Equation (3) to obtain the semantic vector z_i used by the MLP classifier. This design choice does appear in Section 4.2 as the phrase “penultimate layer embeddings,” but we did not explicitly state near the equation that “we set l to the penultimate transformer layer by default,” which understandably caused confusion. In the final version, we will add an explicit clarification around Equation (3) stating that l **is fixed to the penultimate transformer layer of the frozen LLM in all experiments**.

---

### Official Review · Reviewer_kj3J · 2025-11-01

**Soundness:** 1
**Presentation:** 3
**Contribution:** 2
**Rating:** 2
**Confidence:** 4

**Summary:**

This paper introduces GUARD (Generation-time Unlearning via Adaptive Restriction and Detection), a novel training-free framework designed to perform LLM unlearning entirely at inference time, thereby avoiding the catastrophic forgetting and utility degradation common in fine-tuning-based approaches. The proposed system operates via a three-stage pipeline: first, a lightweight MLP classifier identifies if an input prompt pertains to the forget set; second, for positive queries, SBERT-based retrieval identifies the original answer and extracts a corresponding set of "forbidden tokens"; finally, a core contribution of controlled generation dynamically regulates the output during beam search. This control mechanism is a hybrid, combining a Trie-based, token-level hard matching for exact sequence blocking with an SBERT-based soft semantic matching to penalize semantically similar evasions, applying a total penalty to the token's log-likelihood at each decoding step. Extensive experiments across TOFU, MUSE-News, and Harry Potter benchmarks demonstrate that GUARD effectively prevents outputting target information without compromising the model's general capabilities.

**Strengths:**

1. The paper introduces GUARD, a highly original training-free framework that reframes unlearning as an inference-time output regulation problem. Its core technical novelty lies in the synthesis of a Trie-based hard-matching for exact sequence blocking and an SBERT-based soft semantic matching to penalize evasions.

2. The most significant contribution is demonstrating that unlearning can be achieved without parameter modification, thereby completely avoiding the catastrophic forgetting and utility degradation that plague fine-tuning-based methods. Experiments across three benchmark datasets demonstrate that GUARD achieves a superior trade-off between forget quality and knowledge retention.

3. The paper is well-written and easy to follow, supported by high-quality empirical validation. The authors conduct comprehensive baseline comparisons and detailed additional experiments to verify the effectiveness of the proposed modules.

**Weaknesses:**

While this paper introduces a novel and practical approach to unlearning, there are critical weaknesses as below:

1. The primary concern is the substantial computational overhead introduced "at inference time." The GUARD framework requires executing both Trie matching and SBERT semantic similarity checks for every candidate token at every decoding step.
- Appendix G (Table 8) confirms this is a critical bottleneck. Even after an SBERT-caching optimization, the method is slower in batch processing than standard beam search (and was slower pre-caching). This latency increase could be prohibitive for real-world, low-latency applications.
- In this regard, the authors should analyze how inference latency scales as the number of "forbidden tokens" increases (e.g., from 100 to 10,000) to evaluate the method's scalability properly. Additionally, the authors need to compare the proposed method's inference time with other training-free methods.

2. The proposed method is fundamentally a sophisticated output filter, not a true parameter-level "unlearning" mechanism. The knowledge remains fully intact within the model's weights.
- The method blocks specific token sequences and their local semantic neighbors. It is not designed to prevent the model from leaking the same information via deep paraphrasing. It is also incapable of unlearning abstract concepts like bias or misconceptions, which are not tied to specific strings. Even though the authors demonstrate the method's robustness in this paraphrasing scenario, I believe there is still a limitation that prevents it from handling all kinds of paraphrasing in the real world.
- Therefore, I think the authors should frame the work more precisely as "inference-time content suppression". The evaluation should be strengthened by testing against diverse adversarial prompts designed to elicit the core concept through complex rephrasing.

3. Several questions arise from the reported experimental results regarding the baselines.
- NPO (specifically NPO+RT) has been reported in prior work to achieve strong results on the TOFU dataset, particularly in challenging 5% and 10% settings. However, the NPO+RT results in this paper show a significant, unexplained discrepancy from those previously published figures.
- Additionally, the performance of other baseline algorithms appears to be reported as substantially lower than in established literature.
- The authors must thoroughly verify and confirm that these baseline algorithms were reproduced correctly. This includes a review of the experimental setup, hyperparameters, and resulting metrics to ensure a fair and accurate comparison.

**Questions:**

I wrote my major weakness/questions in the main Weakness section. I wrote the remaining questions below.

1. Are the experimental results reported in the tables (e.g., Tables 1, 2, 3) the product of a single experimental run or an average over multiple runs (i.e., multiple seeds)? If they are from a single seed, this limits the assessment of the method's stability.

2. The paper's primary trade-off is sacrificing inference time (Weakness 1) to avoid the pitfalls of training. However, recent works (e.g., [1, 2]) have proposed cost-efficient, training-based unlearning using methods like LoRA. Especially, [2] reports that they achieve strong forget/retain trade-offs (for the TOFU dataset) without incurring any additional inference-time cost.
- What do the authors view as the primary advantage of GUARD when compared to these efficient parameter-lite fine-tuning methods?
- Can the authors provide a compelling argument or experimental comparison to demonstrate that their inference-time approach is superior to (or necessary in addition to) methods like those in [2]?


[1] RWKU: Benchmarking Real-World Knowledge Unlearning for Large Language Models, NeurIPS 2024

[2] Towards robust and cost-efficient knowledge unlearning for large language models, ICLR 2025

---

> ### Author Response · Authors · 2025-11-24
>
> We sincerely thank the reviewer for the time and careful effort devoted to evaluating our work. We deeply appreciate the reviewer’s recognition of the originality of GUARD, its training-free design, and the strength of our empirical validation. Below, we address the reviewer’s concerns and questions in detail.
>
> > **Clarification on Inference-Time Unlearning Efficiency (respone to W1)**
>
> We sincerely thank the reviewer for raising important questions regarding inference-time computational efficiency. The additional overhead introduced during decoding is indeed a key challenge faced by all *training-free unlearning* approaches. Below, we provide a more systematic clarification and supplementary analysis.
>
> **(1) Practical latency impact of GUARD.**
>
> Our profiling results show that the cost of Trie matching is extremely small. Its complexity is O(L) string checking, and the overhead accounts for less than one percent of the total decoding time. Therefore, it is not the primary source of latency. The real contributor to additional cost is the SBERT-based semantic filtering module. However, it is important to emphasize that SBERT is merely a replaceable component within GUARD and is not a core dependency of the framework. It can be substituted with lighter encoders such as MiniLM, GTE, or bge-micro-v2. It can also be compressed via vector quantization, Faiss-based approximate retrieval, or replaced with a small two-to-three-layer Transformer encoder to reduce the computational cost of similarity evaluation. The design of this module is thus flexible and controllable in complexity.
>
> As shown in Appendix G (Table 9), once we enable a simple one-time embedding caching optimization (encode-once-reuse), the inference latency of GUARD (with cache) decreases substantially. **The per-sample latency (mean=417.7 ms) becomes very close to that of standard beam search (383.3 ms), with only about a nine percent difference.** The p95 latency of the two methods is nearly identical. This demonstrates that, after caching, the semantic filtering module does not introduce a meaningful bottleneck for practical deployment.
>
> In real deployment settings, **the filtering logic of GUARD is activated only when a user query triggers a potential unlearning risk. The vast majority of requests go through the standard decoding path. This means the overall average system latency is significantly lower than the worst-case analysis.**  Since the cached version of GUARD already matches the performance of standard beam search, its impact on real-world service latency is minimal.
>
> **(2) Latency impact of forbidden text size**
>
> Following the reviewer’s suggestion, we conducted a systematic experiment evaluating how increasing the size of the forbidden set affects inference latency. All models and inference hyperparameters were kept identical. We varied the size of the forbidden set M from 100 to 10,000. We used Phi-1.5 (including our fine-tuned variant) as the base model, with batch size 128, beam size 7, and max_new_tokens 50. All runs were executed under the same GPU environment. To avoid semantic confounding factors, we randomly sampled M token embeddings from the vocabulary matrix as the forbidden vectors, thereby isolating the true computational overhead associated with increasing M.
>
> | **M (number of forbidden vectors)** | **mean latency** |
> | ----------------------------------- | ---------------- |
> | 100                                 | 417.0 ms         |
> | 500                                 | 415.5 ms         |
> | 1000                                | 439.9 ms         |
> | 2000                                | 429.7 ms         |
> | 5000                                | 427.6 ms         |
> | 10000                               | 441.2 ms         |
>
> The results show that increasing M from 100 to 10,000 causes only marginal fluctuations in latency. Both batch and single-sample inference remain stable and fully controllable. This is because the main computation of GUARD is already completed during the pre-decoding embedding caching stage. Constructing the forbidden set is simply selecting a submatrix from the vocabulary embedding matrix, **which is a lightweight O(M) operation performed once before generation.** Its contribution to overall decoding latency is negligible.

---

> > ### Author Response · Authors · 2025-11-24
> >
> > **(3) Comparison with other training-free methods.**
> >
> > The training-free baselines mentioned in our paper, such as ICUL, output-level filtering, and prompt-based strategies, operate essentially as constraint-based or rule-based approaches. They do not modify the logit layer or perform any vector operations, and thus their runtime is nearly identical to greedy decoding or standard beam search. Table 9 confirms that their inference time can be considered as having nearly zero additional overhead. However, it is important to highlight that this low cost comes at a significant performance trade-off. These methods exhibit very limited forgetting ability across all core evaluation metrics, and often fail to block target knowledge effectively. In other words, **their low computational cost is achieved by sacrificing unlearning effectiveness.** This further illustrates that practical training-free unlearning requires stronger technical mechanisms, which is precisely the motivation behind our design of GUARD.
> >
> > > **Clarification on the Scope and Definition of Inference-Time Unlearning  (response to W2, Q2)**
> >
> > We sincerely thank the reviewer for their thoughtful comments regarding the methodological boundaries and task definition. Regarding the term “content suppression,” we believe this characterization does not fully reflect the goal or capability of our approach. We first clarify that our method is not a parameter-level unlearning technique, but rather belongs to the paradigm of inference-time unlearning. Traditional unlearning frameworks typically rely on gradient updates to modify model parameters. However, in many real-world deployment scenarios, this assumption does not hold. For example:
> >
> > **(1) Large-scale models with tens of billions of parameters make retraining or additional fine-tuning prohibitively expensive.**
> >
> > **(2) In practical applications, models are frequently deployed in a frozen state with weights that cannot be modified.**
> >
> > **(3) Some systems operate under a grey-box assumption, where logits are accessible during inference but model parameters cannot be accessed or altered.**
> >
> > In such settings, parameter-level unlearning is nearly infeasible, whereas **training-free unlearning becomes a realistic and necessary technical pathway.** Therefore, the goal of our method is to intervene during inference when training, updating, or accessing model parameters is impossible, and to block the activation and output of target knowledge so that the model can no longer utilize or expose such information. Importantly, **this inference-time unlearning objective is fully aligned with the community’s widely accepted functional definition of unlearning, namely preventing the model from continuing to use or reveal information that should be forgotten.**[1]  A growing body of recent work also focuses on inference-time unlearning, and this line of research is increasingly recognized within the community [2~4].

---

> > > ### Author Response · Authors · 2025-11-24
> > >
> > > In Appendix B, we have already provided a systematic evaluation of the model’s behavior under diverse rephrasing scenarios, including lexical substitution, syntactic rewriting, structural transformation, and entity masking. The experimental results demonstrate that our method consistently blocks the leakage of target knowledge under these challenging rewriting conditions, and significantly outperforms other training-free baselines. At the same time, we acknowledge that in the real world there may exist infinitely many forms of deep paraphrasing. Even parameter-level unlearning methods cannot theoretically guarantee coverage of all possible transformations. This limitation is a fundamental challenge for the unlearning community and relates more broadly to problems studied in model editing and alignment [5-6]. For this reason, our method focuses on realistically definable factual unlearning and provides the most robust and practical inference-time blocking capability within the enumerable paraphrasing space.
> > >
> > > | **Dataset**    | **FNR_D^Train_ori** | **FNR_D^Test_rephara** | **FNR_D^Test_adv** | **FNR_D^Test_irr** | **FNR_D^Test_jail** |
> > > | -------------- | ------------------- | ---------------------- | ------------------ | ------------------ | ------------------- |
> > > | TOFU (1%)      | 0.0                 | 0.0256                 | 0.0256             | 0.0256             | 0.0                 |
> > > | TOFU (5%)      | 0.0                 | 0.0015                 | 0.0065             | 0.0400             | 0.0025              |
> > > | TOFU (10%)     | 0.0                 | 0.0100                 | 0.0429             | 0.0175             | 0.0049              |
> > > | HP Book        | 0.0                 | -                      | -                  | 0.0                | 0.0                 |
> > > | News (knowmem) | 0.0                 | 0.0100                 | 0.0208             | 0.0392             | 0.0099              |
> > > | News (verbmem) | 0.0                 | -                      | -                  | 0.0                | 0.0                 |
> > >
> > > | **Dataset**    | **FPR_D^Train_ori** | **FPR_D^Test_rephara** | **FPR_D^Test_adv** | **FPR_D^Test_irr** | **FPR_D^Test_jail** | **FPR_D^Test_g** |
> > > | -------------- | ------------------- | ---------------------- | ------------------ | ------------------ | ------------------- | ---------------- |
> > > | TOFU (1%)      | 0.0                 | 0.0002                 | 0.0                | 0.0                | 0.0002              | 0.0004           |
> > > | TOFU (5%)      | 0.0                 | 0.0003                 | 0.0008             | 0.0047             | 0.0003              | 0.0021           |
> > > | TOFU (10%)     | 0.0                 | 0.0011                 | 0.0011             | 0.0013             | 0.0008              | 0.0033           |
> > > | HP Book        | 0.0                 | -                      | -                  | 0.0004             | 0.0002              | 0.0057           |
> > > | News (knowmem) | 0.0                 | 0.0                    | 0.0                | 0.0                | 0.0100              | 0.0056           |
> > > | News (verbmem) | 0.0                 | -                      | -                  | 0.0                | 0.0                 | 0.0001           |
> > >
> > > >  **Clarification on Baseline Reproduction Results (response to W3)**
> > >
> > > We thank the reviewer for their attention to the reproduced baseline results. First, as detailed in Appendix E, we report the full experimental settings for all baseline algorithms, including training steps, hyperparameter configurations, optimizer settings, and evaluation protocols. Our overall pipeline strictly follows the configurations described in the original paper as well as those used in prior related work [7–8].
> > >
> > > Additionally, several independent studies [8–9] report NPO and other baseline metrics that fall within the same numerical range as our reproduced results. This further indicates that our implementation lies within a reasonable and expected reproducibility range, rather than being abnormally low.
> > >
> > > >  **On the Use of Single-Seed Results (response to Q1)**
> > >
> > > We thank the reviewer for their concern regarding the stability of the reported results. In our main experiments, we follow the same setup adopted by a large body of widely accepted unlearning literature, where results are reported from a **single experimental run** (e.g., [7-9]). We adhere to these standard practices to ensure comparability of baselines and consistency of experimental settings.

---

> > > > ### Author Response · Authors · 2025-11-24
> > > >
> > > > **Referneces:**
> > > >
> > > > [1] Yao Y, Xu X, Liu Y. Large language model unlearning[J]. Advances in Neural Information Processing Systems, 2024, 37: 105425-105475.
> > > >
> > > > [2] Pawelczyk M, Neel S, Lakkaraju H. In-Context Unlearning: Language Models as Few-Shot Unlearners[C]//International Conference on Machine Learning. PMLR, 2024: 40034-40050.
> > > >
> > > > [3] Thaker P, Maurya Y, Hu S, et al. Guardrail baselines for unlearning in llms[J]. arXiv preprint arXiv:2403.03329, 2024.
> > > >
> > > > [4] Liu C, Wang Y, Flanigan J, et al. Large language model unlearning via embedding-corrupted prompts[J]. Advances in Neural Information Processing Systems, 2024, 37: 118198-118266.
> > > >
> > > > [5] Lynch A, Guo P, Ewart A, et al. Eight methods to evaluate robust unlearning in llms[J]. arXiv preprint arXiv:2402.16835, 2024.
> > > >
> > > > [6] Barez F, Fu T, Prabhu A, et al. Open problems in machine unlearning for ai safety[J]. arXiv preprint arXiv:2501.04952, 2025.
> > > >
> > > > [7] Maini P, Feng Z, Schwarzschild A, et al. Tofu: A task of fictitious unlearning for llms[J]. arXiv preprint arXiv:2401.06121, 2024.
> > > >
> > > > [8] Wang Y, Wei J, Liu C Y, et al. LLM Unlearning via Loss Adjustment with Only Forget Data[C]//The Thirteenth International Conference on Learning Representations.
> > > >
> > > > [9] Pang Z, Zheng H, Deng Z, et al. Label Smoothing Improves Gradient Ascent in LLM Unlearning[J]. arXiv preprint arXiv:2510.22376, 2025.

---

### Note · Authors · 2025-12-02

I have read and agree with the venue's withdrawal policy on behalf of myself and my co-authors.